# Using Laminar Nanoclays for Phycocyanin and Phycoerythrin Stabilization as New Natural Hybrid Pigments from Microalgae Extraction

**Bàrbara Micó-Vicent [1,*], Esther Perales Romero [2], Ruperto Bermejo [3], Jorge Jordán-Núñez [1], Valentín Viqueira [2] and Jorge Pérez [2]**

[1] Departamento de Ingeniería Gráfica, Universitat Politècnica de València, 03801 Alcoy, Spain; jorjornu@upv.es
[2] Departamento de Óptica Farmacología y Anatomía, Universidad de Alicante, 03690 Alicante, Spain; esther.perales@ua.es (E.P.R.); valentin.viqueira@ua.es (V.V.); jorge.perez@ua.es (J.P.)
[3] Department of Physical and Analytical Chemistry, Higher Engineering Polytechnic School of Linares, University of Jaen, 23700 Linares, Spain; rbermejo@ujaen.es
[*] Correspondence: barmivi@upv.es

**Featured Application: Hybrid pigments from synthetic or natural nanoclays confer organic compounds stabilization as the main advantage. Nanoclays adsorb microalgae pigments, which improves the optical, mechanical and thermal properties of adsorbed organic compounds and the material in applications. This work provides the stabilization of phycocyanin and phycoethrin proteins, which can be used in several applications including coatings, polymer additives (i.e., 3D printing), cosmetics, and textile industries.**

**Abstract:** C-Phycocyanin (PC) and B-phycoerythrin (PE) are light-harvesting water-soluble phycobiliproteins from microalgae that belong mainly to the cyanobaceria and rhodhophytes families. Different methods have been developed for PC and PE extraction and purification from microalgae, and offer a high potential for their use as additives in sectors such as food and cosmetics. However, the main limitations of using these dyes are the sensitivity of their environmental factors, such as light fastness, temperature, and pH. We successfully employed safe lamellar nanoclays such as montmorillonite (M) and Laponite (L) for phycobiliproteins stabilization, as we did before with other natural dyes. We obtained a wide color gamut from blues to pink by combining four different factors under synthesis conditions: three dye concentrations; two laminar nanoclay sizes; a two nanoclay surface modifiers combination with cetylpyridinium bromide (CPB) and a coupling agent (3-Aminopropyl) triethoxysilane. The experimental conditions were defined according to a multilevel factorial design of experiment (DOE) to study the factors interacting in the final hybrid pigment characteristics. In both M and L, the d001 distance (nm) increased due to PC and PE adsorption. The best conditions to increase the basal space depend on the nanoclay structure, and it is better to use the surfactant for M, and silane modification for L. In addition, optical and thermal PE and PC properties significantly improved. We show the optimal synthesis conditions to increase PC and PE adsorption using the high dye concentration, with surfactant and silane depending on the nanoclay. The hybrid pigments from these phycobiliproteins offer the opportunity to perform several industrial applications, including in polymer additives, cosmetics, and packaging.

**Keywords:** phycocyanin; phycoerythrin; stabilization; statistical design of experiments; montmorillonite; laponite; hybrid pigments

## 1. Introduction

Algal resources are not widely eaten, but are considered promising agricultural resource because they contain considerable amounts of phenolic substances, carotenoids and antioxidative proteins. Pigments that produce microorganisms and microalgae are quite

common in nature, but there is still a long way to go from the laboratory scale to the market place. Fungi, bacteria and microalgae provide carotenoids or phycocyanin at the industrial level [1,2]. Phycocyanin (PC) is one of the main pigments of the alga *Arthrospira platensis*, a protein from the phycobiliprotein family characterized by its intense blue color and a nutraceutical compound with antioxidant activity. The PC food grade is generally recognized as safe (GRAS) food and employed as a dye in food product formulations, and also in cosmetics, alcoholic beverages, biotechnology and medicine (drugs) [3]. In this context, PC has been mainly studied as a natural colorant for dairy products, yogurts and other beverages [4–9]. However, the application of this pigment–protein complex compound is limited, mostly due to its lack of stability to heat, light and acidic conditions. When applied as a natural food colorant, degradation of color, concentration and antioxidant activity often occurs due to high temperatures during food processing. PC gradually changes from light blue to faint blue, and this transformation is a disadvantage in industry because of the resulting unattractive hue [10]. Several techniques have been used to prevent thermal PC degradation, such as the addition of a stabilizer, pH adjustment and encapsulation [11].

Phycoerythrin (PE) is a major light-harvesting pigment of red algae, and the red algae *Porphyridium cruentum* is an important source of this biomolecule and can be used as a natural dye in several applications for its unique pink color, high absorption coefficient and marked fluorescence properties, such as high quantum yield and marked Stokes' shift [12]. This pigment is a valuable candidate for the design and characterization of light-sensing elements in biosensors [13], and can be used in cosmetics or as a food colorant to replace synthetic dyes [14–16] It possesses properties to protect against physiological changes under oxidative stress with strong anti-aging effects [17,18].

However, industrial applications from biliproteins are limited by their chemico-thermal stabilization. Acid pH and high temperatures are factors that affect spectral PE properties, which make their control during applications difficult. Some applications require solubilization in nonpolar solvents. The best storage conditions have been determined as −20 ºC using sodium phosphate buffer (20 mM, pH 7.1) as the dye solvent [19].

In light of the above, natural dye applications from PC or PE have limited by their colorfastness. Exposure to light, being temperature-sensitive or spectral changes under different pH conditions all limit their use. Employing nanoclays as inorganic adsorbents has proved useful for the stabilization of different organic compounds with optical and thermal properties, which different applications require. Several nanoclay classifications exist depending on their structure or origin. Montmorillonite (M) is the most abundant mineral in the smectites group. It is a natural, abundant and cheap clay that offers efficient adsorbance. Its structure is formed by two layers, a tetrahedric silicon one and an octahedric aluminum oxide one, which create a 2:1 diactahedral layer characterized by having a wide dehydroxylation temperature range of 500–700 °C [20,21]. The porosity and ion adsorption of MMT are characteristics that have been thoroughly studied by scientists. The purpose of many research works has been to achieve wider basal spacing [22]. Much interest has been shown in the M exchange control in recent decades. With MMTAP, this control takes place by variation in pH, and also by the interlaminar space in which ion exchange occurs [23].

Synthetic laponite (L) ($Si_8[Mg_{5.5}Li_{0.4}H_{4.0}O_{24}]^{0.7}-Na^{0.7+}$) is a natural 2D clay disc-shaped silicate whose dimensions are approximately 1 nm thick with a diameter of 25 nm [24,25]. L is a natural inorganic stratified silicate element that is often used to improve the rheology of different water-based products [26,27]. Its capacity to react with water-based components is excellent, and its viscosity develops when such products are incorporated [28]. Some studies have demonstrated the adsorption capacity of LAP-based hydrogels [29]. Several factors with a proven effect on nanoclay adsorption capacities have been studied, such as temperature, stirring speed, ion strength, and use of surfactants or silane compounds [30].

The use of hybrids nanopigments from inorganic hosts such as laminar nanoclays and natural dyes extracted from agrowaste has been successful in advanced applications such as 3D-printing technologies. When hybrid nanopigments are exfoliated into a polymer

matrix, the optical, thermal, and mechanical properties of that matrix improve, which provides original dye application possibilities [31].

As a green chemistry trend, recent studies have worked on the stabilization of natural dyes, or synthetic similar synthetic cations, onto inorganic substrates, which results in the formation of hybrid pigments. For example, the use of flavylium cations such as synthetic analogs to anthocyanins demonstrates that laminar nanoclays, e.g., M and L, can efficiently adsorb these cations to create hybrid pigments with improved properties [32]. To overcome the poor stability of natural lutein to environmental factors, hydrotalcite has been incorporated by a green mechanical grinding process. The thermal decomposition of lutein/LDH has been significantly improved by the chemical interaction [33]. Vermiculite improves hybrids pigment using alizarin cations to improve elastomer flammability properties [34]. Aluminum-magnesium hydroxycarbonate (LH) modified with carmic acid improves Polymmer composite properties by reducing polymer matrix flammability. CA stabilization on LH leads to pigments with excellent resistance to acetone and water [35]. In conclusion, clay minerals are promising carriers for enhancing the stability of plant pigments. An optimal promising strategy is possible to enhance the stability of natural pigments against various chemical reagents, high temperature and visible light irradiation after incorporating natural or synthetic clay minerals [36].

Several research works still work on natural dye adsorption optimization by nanoclay structures [37]. We selected the natural proteins extracted from microalgae adsorption (PC and PE) as a host in the nanoclay compounds. We also compared the nanoclay structure, silane and surfactant interactions in PC and PE adsorption [38]. In addition, we compared the optical properties of the hybrid pigments synthesized under different conditions to know the optimal combination to achieve a wide color gamut with both proteins. In this work, we employed natural M and synthetic L nanoclays for PC and PE stabilization. We applied statistical design of experiment (DOE) to combine different factors during the synthesis process to find the optimal conditions for maximum dye adsorption. We aimed to study the factors that influenced hybrid pigment properties, such as color gamut, thermal fastness or total solar reflectance (%). The studied responses will be the main benefit of the hybrid pigments generated from microalga sources.

## 2. Materials and Methods

### 2.1. Synthesis

In this work, we used laminar nancolays from different sources and size particles: MMT with trade name Gel white®, and Laponite® (LAP) with trade name XLG. The chemical composition of MMT is 80% $SiO_2$, 13% $Al_2O_3$ and 3% $Fe_2O_3$ (molecular formula: $Al_2H_2O_{12}Si_4$) [39]. The cation exchange capacity (CEC) of M is approximately 90–145 $cmol \cdot kg^{-1}$. Synthetic L ($Si_8[Mg_{5.5}Li_{0.4}H_{4.0}O_{24}]_{0.7} - Na_{0.7}^+$) is a 2D clay disc-shaped silicate that is approximately 1 nm thick. Its diameter is 25 nm. The adsorption capacity of L-based hydrogels and their CEC is 74 $cmol \cdot kg^{-1}$ [40]. They were supplied by from BYK Additives Ltd. (Wesel, Germany), and Rockwood, (Widness, UK), respectively. First we attempted to also use hydrotalcite nanoclay as we did before with other natural dyes [41]. However, this was not possible due to the nanoclay pH in dispersion, not even with the calcinated one. pH remained above 4 and it could not be lowered with $HCl^-$ as we did with M or L. Both PC and PE lost their color under these pH conditions and only hydrotalcite optical properties remained. This was why we finally selected the M and L clays.

As nanoclay organic modifiers, we employed surfactant cetylpyridinium bromide (SURF) $C_{21}H_{38}BrN \cdot 6H_2O$, 384.44 g/mol, purity 98%, and silane compound (SIL) 3-aminopropyltriethoxysilane, $H_2N(CH_2)_3Si(OCH_3)_3$, 179.29 g/mol, purity 99%. The pH variation during the synthesis process was controlled by chloridric acid $HCl^-$ (37%). All these products were supplied by Sigma-Aldrich (Madrid, Spain).

Both PC and PE were extracted and purified as in our previous works using buffer phosphate solutions [42,43]. The structure of biliproteins depends on the environment in which they are found (physical state, pH, ionic strength, etc.), and can be a complex mixture

of trimers, hexamers or monomers. Figure 1 depicts as an example of the 3D structure and its dimensions for PC and PE in the hexameric state. The hybrid synthesis process via water adsorption under stirring conditions has also been followed after taking into the account the incorporation time of additives according to our former works [38]. In this study, both nanoclays were dispersed at 1800 rpm for 20 h. The L dispersions were prepared at 18 g·L$^{-1}$ to avoid gel formation in distilled water, while the M dispersions were prepared at 33 g·L$^{-1}$. In both nanoclay dispersions, pH was adjusted to 4 using 37% HCl$^{-}$ [44,45]. Dye solutions (PC and PE) were prepared in buffer phosphate at [250 mM] and were added to the nanoclay dispersions at three concentrations over clay mass: 0.5%, 1% and 3%. The surface modifiers concentration fell within a 0–1% range over nanoclay mass. The solvent separation and drying process by lyophilization were carried out as in our previous works [46].

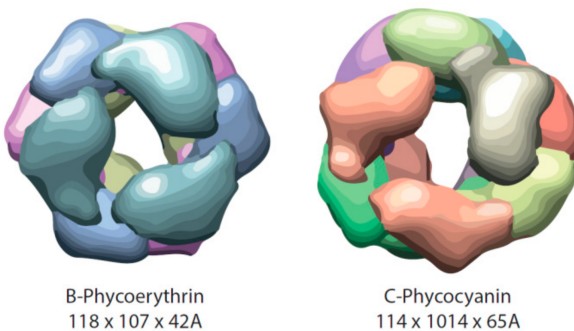

B-Phycoerythrin
118 x 107 x 42A

C-Phycocyanin
114 x 1014 x 65A

**Figure 1.** 3D structure for the biliproteins used in this work: C-Phycocyanin and B-phycoerythrin in the hexameric state.

The experimental conditions were defined using a multilevel factorial DOE by selecting only 16 from the multifactorial $3^1 \cdot 2^1$ experiment conditions according to the D-Optimization criteria (Table 1). We included the dye concentration at three levels, namely the nanoclay origin, and the silane or surfactant concentration, as synthesis factors to control synthesis performance and hybrid pigment properties. Figure 1 shows the images from the different synthesis steps with both the PC and PE dyes and fine powders obtained under distinct synthesis conditions.

**Table 1.** Experimental conditions for the synthesis of both the PC and PE hybrid pigments from the multifactorial $3^1 \cdot 2^1$ DOE (D-Optimization criteria).

| REF. | | CONC [a] | CLAY [b] | SURF [c] | SIL [d] |
|---|---|---|---|---|---|
| nPC.1 | nPE.1 | 3 | M | 0.15 | 0.15 |
| nPC.2 | nPE.2 | 3 | M | 0 | 0 |
| nPC.3 | nPE.3 | 1 | L | 0.15 | 0 |
| nPC.4 | nPE.4 | 0.5 | L | 0.15 | 0.15 |
| nPC.5 | nPE.5 | 3 | L | 0.15 | 0 |
| nPC.6 | nPE.6 | 0.5 | L | 0 | 0.15 |
| nPC.7 | nPE.7 | 3 | L | 0 | 0 |
| nPC.8 | nPE.8 | 3 | L | 0.15 | 0.15 |
| nPC.9 | nPE.9 | 1 | M | 0.15 | 0.15 |
| nPC.10 | nPE.10 | 3 | L | 0 | 0.15 |
| nPC.11 | nPE.11 | 1 | L | 0.15 | 0.15 |
| nPC.12 | nPE.12 | 0.5 | M | 0 | 0.15 |
| nPC.13 | nPE.13 | 0.5 | L | 0 | 0 |
| nPC.14 | nPE.14 | 3 | M | 0 | 0.15 |
| nPC.15 | nPE.15 | 0.5 | M | 0 | 0 |
| nPC.16 | nPE.16 | 0.5 | M | 0.15 | 0 |

[a] PE-concentration; ml of matter solution, [b] Clay structure; montmorillonite (M) or laponite (L). [c] Surfactant concentration (g), [d] Silane concentration (mL).

## 2.2. Characterization

The determination of the amount of dye adsorbed by the nanoclay system allowed us to define the process' synthesis performance. For this purpose, a UV–Vis transmission spectrophotometer (JASCO V650, Easton, MD, USA) was utilized to measure the dye absorbance (%) in the separated supernatants. Then the amount of dye adsorbed by nanoclays was calculated as a percentage of the initial concentration in the exchange step. This parameter was employed as a response to be minimized in the DOE analysis.

A transmission spectrophotometer with an integration sphere, the double UV-Vis/NIR Jasco V-670, was used to measure the spectral absorbance of all the synthesized hybrid pigments. Color properties such as lightness $L_{ab}*$, hue $H_{ab}*$ and saturation $C_{ab}*$ were calculated and represented in the CIELAB color space to check the color gamut used with the D65 illuminant and the CIE-1964 standard observer. Total solar reflectance (TSR%) was calculated from all the hybrid pigments [47].

XRD Bruker D8-Advance equipment (Bruker, Billerica, MA, USA), with a Göebel mirror (power: 3000 W, voltage: 20–60 kV and current: 5–80 mA), was employed. Measurements were taken in an oxidant atmosphere at an angular speed of $1°/\text{min}$, STEP $0.05°$, and an angular scan of $2.7–70°$. XRD patterns were obtained to observe variations in the basal space on the layers from the different nanoclays due to interactions with the natural dye and modifiers.

Finally, to characterize the thermal stability of all the synthesized pigments and to make comparisons with the original extracted PC and PE, a thermogravimetric analyzer TGA/SDTA 851 (Mettler-Toledo Inc., Columbus, OH, USA) was utilized at a temperature ramp of $5\,°\text{C/min}$ within the $20–900\,°\text{C}$ range with oxidant medium $N_2:O_2$ (4:1).

## 3. Results and Discussion

The results obtained from the separated supernatants and the powders from the synthesized hybrid pigments were used as responses in the statistical analysis by the DOE results to describe the influence of factors on each result/response.

## 3.1. Adsorption

The first analysis involved taking the measurement of the dye concentration in the separated supernatants. This is an indirect measurement of a dye bound with nanoclays that indicates synthesis performance. The first step was to obtain the calibration line for each dye in water solution. We added different concentrations of the most concentrated buffer phosphate solution to distilled water and obtained the calibration lines seen in Figure 2. Then we made up the supernatants to a known volume and took measurements with the spectrophotometer to know the dye concentration in the supernatant separated after the exchange process. Once the dye concentration added to each nanoclay dispersion was known, we calculated the dye percentage bound to the nanoclay as the adsorption percentage (Ads (%)). As Table 2 shows, adsorption (%) moved from 7% to 100% depending on the synthesis conditions. The influences of factors, such as their interactions, can be explained and were found in the analysis of variance (ANOVA) (Table 3). The significant factors observed for the Ads (%) response were the clay origin (CLAY), the silane concentration (SIL) and the silane concentration or silane clay interactions ($p$-Values lower than the significant level).

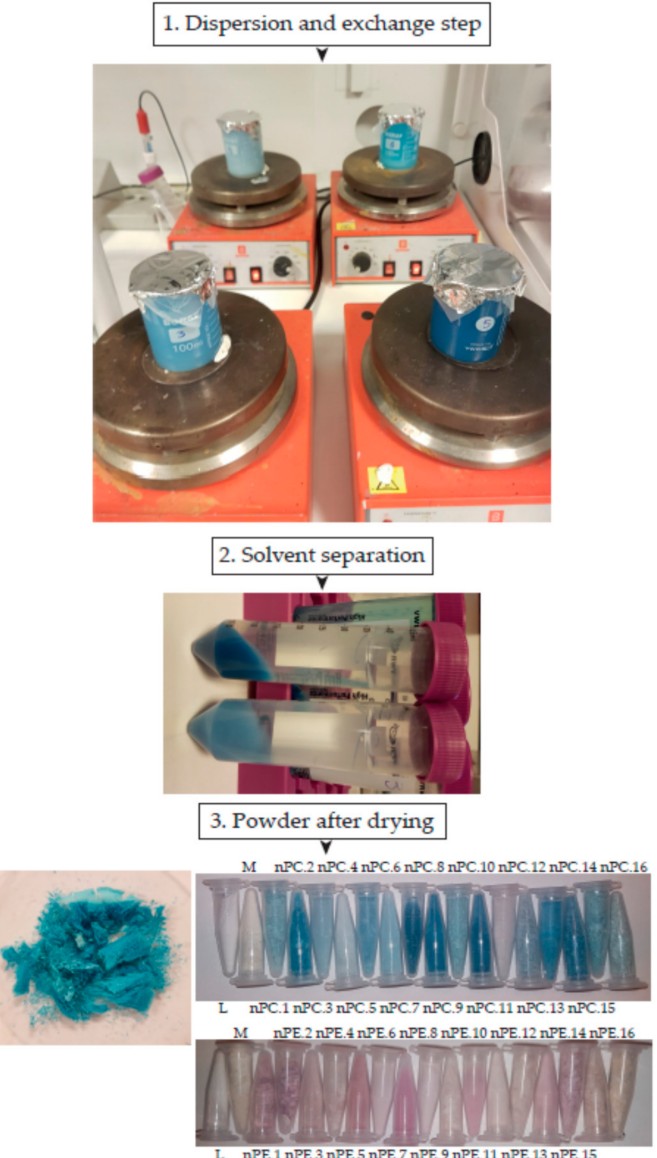

**Figure 2.** Pictures of the different synthesis steps and the results in the powder format for all the hybrid pigments with dyes PC and PE.

As seen in Figure 3, the maximum adsorption in both dyes was found with the maximum added dye concentration using silane after dye addition. However, the presence of silane lowered the dye adsorption in PE, but increased adsorption in PC. Depending on the dye structure, the nanoclay modification with silane helped dye adsorption or inhibited adsorption in the laminar nanoclay structure. Both natural dyes are photosynthetic proteins and have prosthetic groups called bilins, which are responsible for color response. Classification as PC or PE depends on their absorption and emission; PC (610–620 nm) and PE (540–570 nm). Their sizes are also affected by their configuration. PE has a heavier molecular weight than PC: 240,000 Da vs. 36,700 Da. These configuration and volume differences explain the wide adsorption variations noted during PE synthesis compared to the adsorption variations for PC. PC molecules adsorption in nanoclay structures implied less energy than PE molecules.

**Table 2.** Adsorption (%) from the initial PC and PE concentrations under different synthesis conditions (DOE) using supernatant measurements (UV-VIS).

| REF. | Ads (%) [a] | REF. | Ads (%) [b] |
|------|------|------|------|
| nPE.1 | 67.15 | nPC.1 | 96.87 |
| nPE.2 | 78.16 | nPC.2 | 95.23 |
| nPE.3 | 18.27 | nPC.3 | 99.30 |
| nPE.4 | 7.90 | nPC.4 | 98.44 |
| nPE.5 | 76.18 | nPC.5 | 100.00 |
| nPE.6 | 29.35 | nPC.6 | 97.56 |
| nPE.7 | 31.65 | nPC.7 | 99.91 |
| nPE.8 | 80.54 | nPC.8 | 100.00 |
| nPE.9 | 28.80 | nPC.9 | 100.00 |
| nPE.10 | 39.42 | nPC.10 | 100.00 |
| nPE.11 | 16.71 | nPC.11 | 92.62 |
| nPE.12 | 8.76 | nPC.12 | 100.00 |
| nPE.13 | 7.54 | nPC.13 | 88.52 |
| nPE.14 | 63.80 | nPC.14 | 99.16 |
| nPE.15 | 15.87 | nPC.15 | 71.14 |
| nPE.16 | 14.52 | nPC.16 | 76.90 |

[a] sample standard deviation: 27.25, [b] sample standard deviation: 8.76.

**Table 3.** Analysis of variance from PC and PE adsorptions (%) under DOE conditions.

| Source | [a] Sum Sq. | [b] d.f. | [c] Mean Sq. | F-Ratio | *p*-Value |
|--------|---------|------|----------|---------|---------|
| | | | **PC** | | |
| A:CONC | 208.407 | 1 | 208.407 | 14.85 | 0.0063 |
| B:CLAY | 88.1283 | 1 | 88.1283 | 6.28 | 0.0406 |
| C:SURF | 0.153236 | 1 | 0.153236 | 0.01 | 0.9197 |
| D:SIL | 249.591 | 1 | 249.591 | 17.78 | 0.004 |
| AB | 9.88922 | 1 | 9.88922 | 0.7 | 0.429 |
| AD | 166.704 | 1 | 166.704 | 11.88 | 0.0107 |
| BD | 113.902 | 1 | 113.902 | 8.12 | 0.0247 |
| CD | 13.9593 | 1 | 13.9593 | 0.99 | 0.3518 |
| Error total | 98.2455 | 7 | 14.0351 | | |
| Total (corr.) | 1150.21 | 15 | | | |
| R$^2$: 78.20% | | | | | |
| | | | **PE** | | |
| A:CONC | 8355.89 | 1 | 8355.89 | 66.75 | 0.0001 |
| B:CLAY | 392.87 | 1 | 392.87 | 3.14 | 0.1198 |
| C:SURF | 222.976 | 1 | 222.976 | 1.78 | 0.2238 |
| D:SIL | 70.5179 | 1 | 70.5179 | 0.56 | 0.4774 |
| AA | 509.76 | 1 | 509.76 | 4.07 | 0.0834 |
| AB | 395.824 | 1 | 395.824 | 3.16 | 0.1186 |
| AC | 935.866 | 1 | 935.866 | 7.48 | 0.0292 |
| BD | 450.666 | 1 | 450.666 | 3.6 | 0.0996 |
| Error total | 876.325 | 7 | 125.189 | | |
| Total (corr.) | 11,134.9 | 15 | | | |
| R$^2$: 90.44% | | | | | |

[a] Sum of Squares, [b] Degrees of freedom, [c] Mean of Squares.

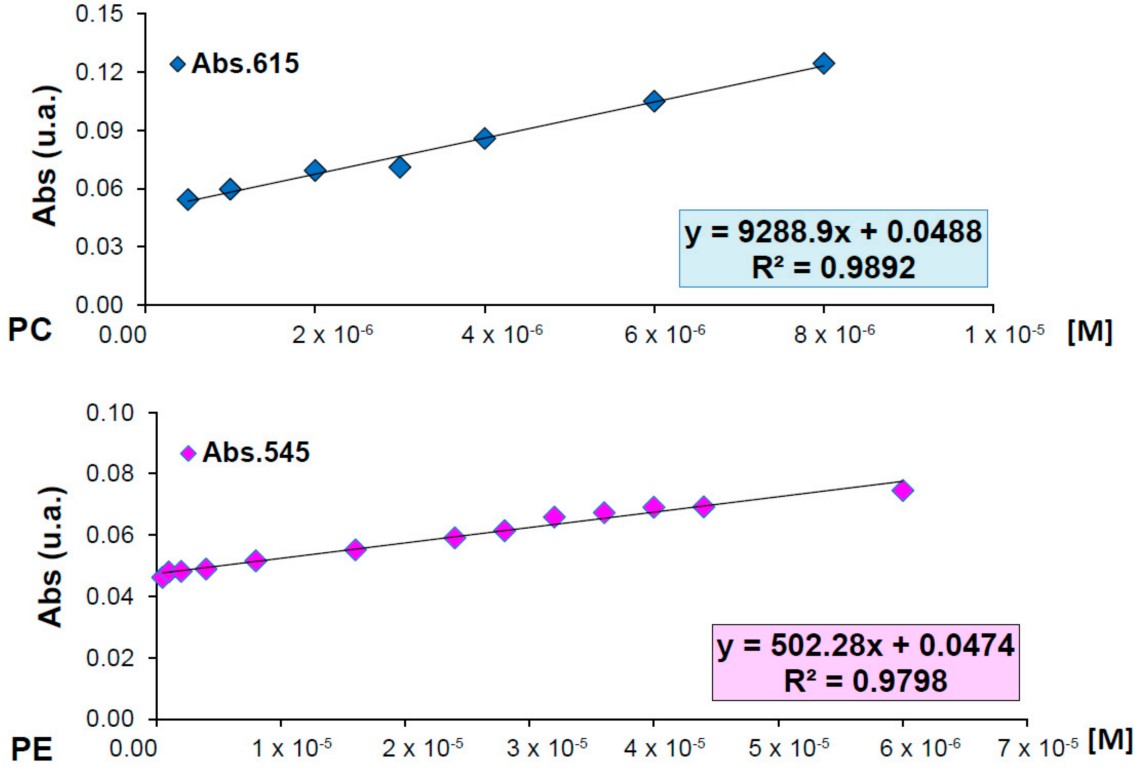

**Figure 3.** Calibration lines from dyes PC ($\lambda$ 615 nm) and PE ($\lambda$ 545 nm) in water dilutions from the initial buffer phosphate concentration.

### 3.2. Color Characterization

Color properties, such as lightness $L_{ab}^*$, hue $h_{ab}^*$ and Chroma $C_{ab}^*$, were calculated. They are represented in the CIELAB color space to compare the color gamut from both hybrid pigments PC and PE. Once again, Table 4 shows how the different applied conditions changed the color perception of the hybrid pigments. We did another ANOVA calculation to know the factors that influenced this variance.

**Table 4.** $L^*$ and $C_{ab}^*$ values for the PE and PC hybrid pigments under different synthesis conditions [1–16] using the D65 illuminant and the CIE-1931 XYZ standard observer.

| REF. | $L^*$ | $C_{ab}^*$ | REF. | $L^*$ | $C_{ab}^*$ |
|---|---|---|---|---|---|
| nPC.1 | 69.77 | 11.08 | nPE.1 | 74.37 | 7.54 |
| nPC.2 | 58.16 | 15.05 | nPE.2 | 71.57 | 6.81 |
| nPC.3 | 68.59 | 12.96 | nPE.3 | 76.25 | 4.96 |
| nPC.4 | 76.13 | 9.23 | nPE.4 | 80.55 | 2.58 |
| nPC.5 | 62.18 | 13.69 | nPE.5 | 65.90 | 9.20 |
| nPC.6 | 76.02 | 13.47 | nPE.6 | 81.17 | 4.65 |
| nPC.7 | 53.05 | 10.62 | nPE.7 | 65.74 | 12.63 |
| nPC.8 | 55.41 | 14.22 | nPE.8 | 83.92 | 4.07 |
| nPC.9 | 70.42 | 11.13 | nPE.9 | 82.44 | 3.68 |
| nPC.10 | 55.87 | 13.79 | nPE.10 | 71.62 | 9.85 |
| nPC.11 | 71.37 | 6.54 | nPE.11 | 79.94 | 4.64 |
| nPC.12 | 69.27 | 8.78 | nPE.12 | 78.85 | 5.02 |
| nPC.13 | 62.34 | 15.04 | nPE.13 | 73.01 | 7.44 |
| nPC.14 | 59.30 | 13.33 | nPE.14 | 71.37 | 7.62 |
| nPC.15 | 73.14 | 11.52 | nPE.15 | 77.49 | 4.44 |
| nPC.16 | 74.63 | 7.09 | nPE.16 | 77.72 | 7.90 |

Figure 4 depicts the color attribute differences in the hybrid pigments from PE. The nPE.16 samples displayed a yellowish hue, while the other samples were reddish, or even bluish, such as nPE.7. Chroma and lightness showed significant differences; for example, nPE.7 or nPE.5 were darker and more chromatic. However, none of these samples represented the optimal conditions for achieving the most varied color gamut. ANOVA calculations had to be done to understand the factors that influenced the final color perception. Figure 5 represents the interaction factors that were significant in the ANOVA calculation.

All the analyzed factors were significant for the chroma or lightness of PC or PE. However, significant interactions depend on PC or PE and perception attribution $C_{ab}^*$ or $L^*$. Figures 5–8 show that, to gain a wide color gamut from a starting point with the lowest lightness and the highest chroma values in PC, the high dye concentration should be used with the L nanoclay without employing surfactant and silane additives. The M clay results were more chromatic at a high concentration than the L blue pigments (PC). However in the pink pigments (PE), the most chromatic samples were achieved with L with no surfactant or silane additives. In these samples, the less chromatic and lighter samples were found at the medium concentration level (1%), with worse results than at the lowest concentration. This was probably due to the agglomeration phenomenon in the nanoclay structure at this concentration. Future applications must take into account either the interactions that occurred between the dye concentration and clay origin or the additives concentration to control both color gamut and color perception. The dye concentration showed nonlinear behavior in the color response, which must also be considered.

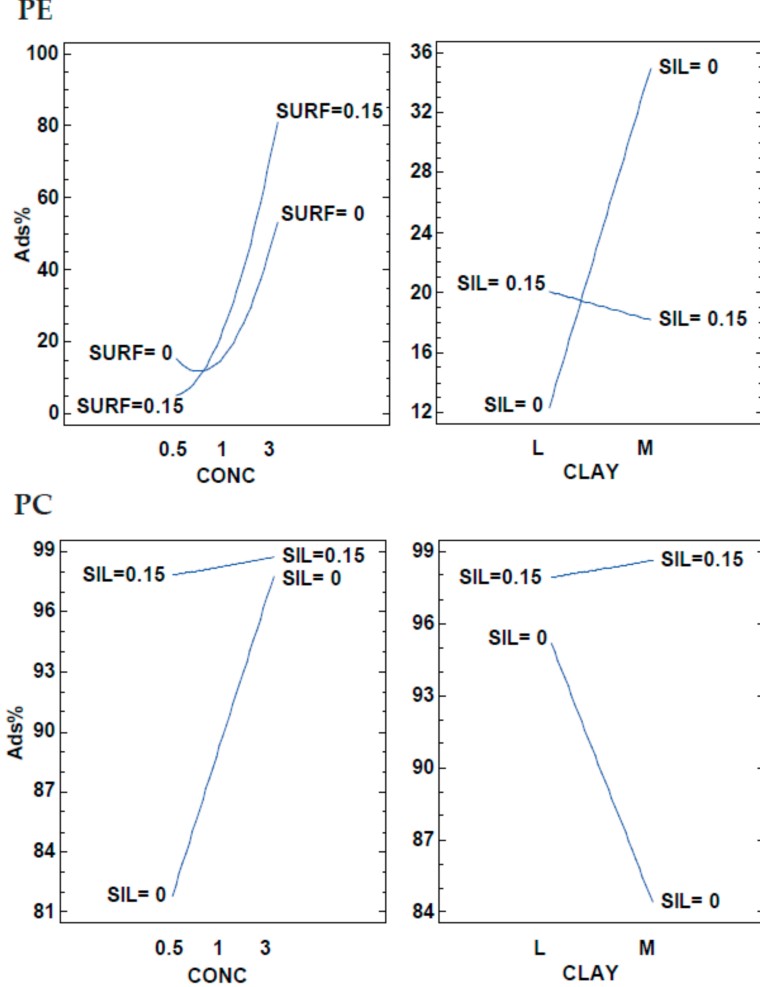

**Figure 4.** Interactions plot of the PE and PC Ads (%) responses under DOE conditions.

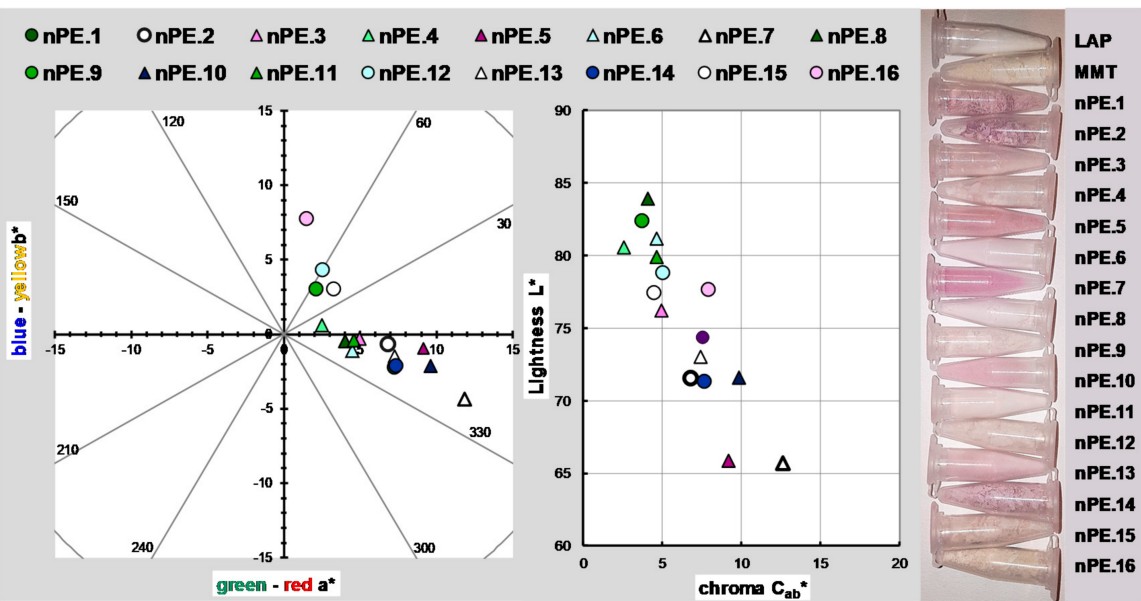

**Figure 5.** Graphic CIELAB plots for the PE hybrid pigments under different synthesis conditions [nPE.1–nPE.16] using the D65 illuminant and the CIE-1931 XYZ standard observer. **Left**: CIE-a* b* color diagram; **right**: CIE-$C_{ab}$* L* color chart.

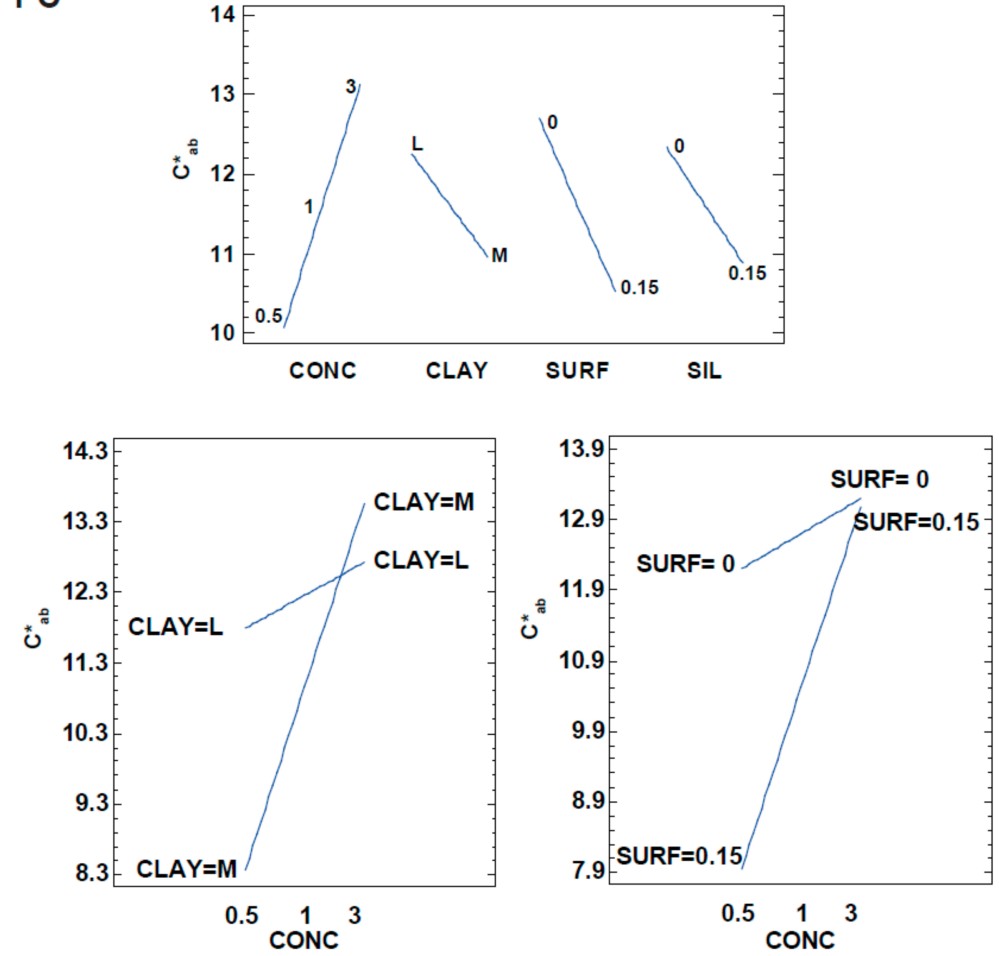

**Figure 6.** Main effects and interaction plots from the $C_{ab}$* response in the PC pigments.

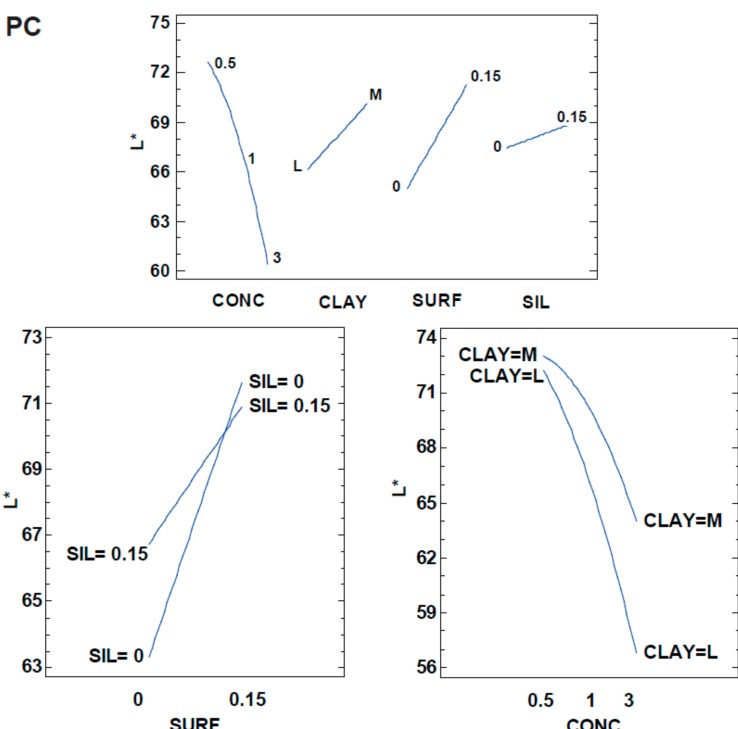

**Figure 7.** Main effects and interactions plots from the L* response in the PC pigments.

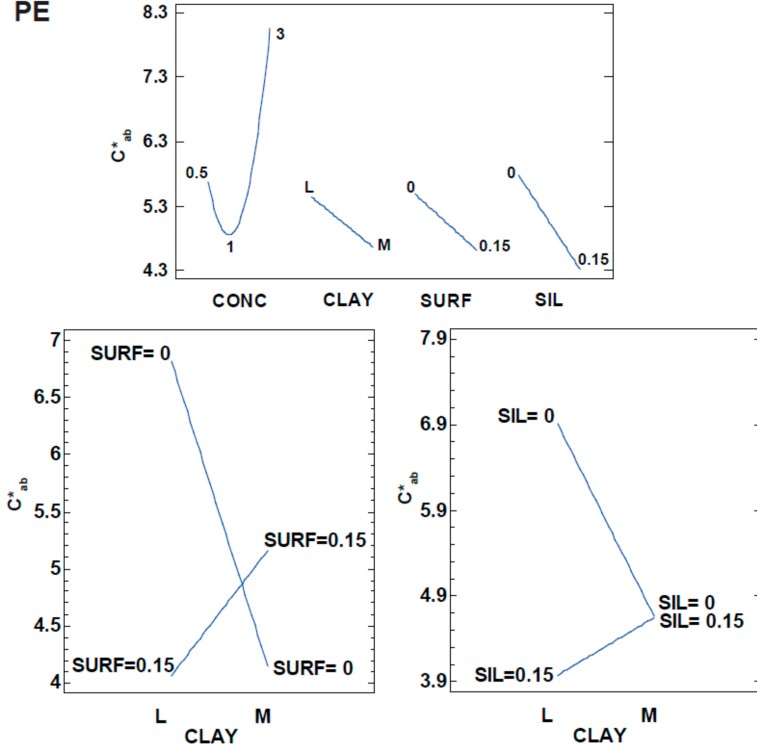

**Figure 8.** Main effects and interactions plots from the $C_{ab}^*$ response in the PE pigments.

### 3.3. Thermogravimetrical Analysis DTA

Thermal analyses were performed using the mass lost by temperature range and the derivate for curves to compare the maximum degradation peaks. Figure 9 shows the main peaks from the original raw materials, namely nanoclays M and L, with a first mass loss at between 60–64 °C. This loss corresponds to the elimination of surface-absorbed water

(free water) and interlayer water [48,49]. PC and PE had different thermal degradation peaks, while PE started degradation at 145 °C and PC at 83 °C, which means that their temperature stability is remarkable [50].

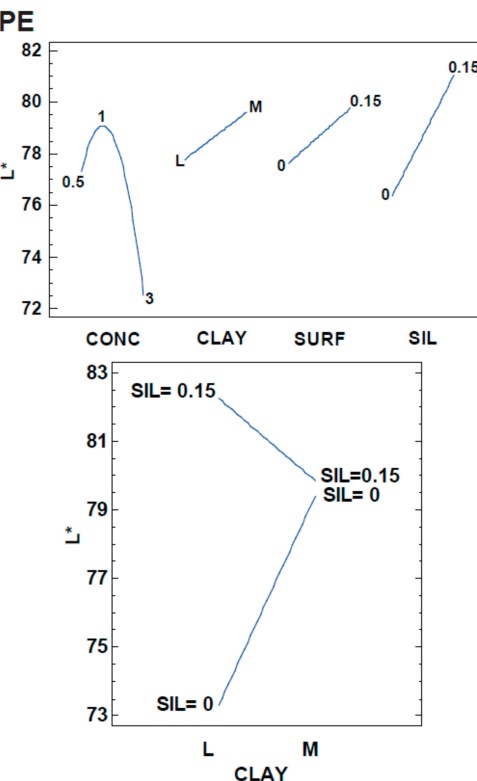

**Figure 9.** Main effects and interactions plots from the L* response in the PC pigments.

Figure 10 shows DTA diagrams from the original components; montmorillonite (M), laponite (L) phycocyanin (PC) and phycoerythrin (PE) (Table S1: TGA_PE_supplementary 1, Table S2: TGA_PE_supplementary 2). Figure 11 shows the thermal stabilization of the PC dye when bound to both nanoclays, which increased the degradation temperature by 5° in all the hybrid samples. In the same samples, the mass loss due to the water separation from both laminar nanoclays was significantly lower than in the original nanoclays due to the cation substitution by the PC dye. The same phenomenon occurred with the hybrid pigments from the PE dye (Figure 12). In all 16 samples, the cation and water displacements from the basal space were reflected in the decreasing mass loss between 25–100 °C. However, in these samples, the water extraction from nanoclays enabled thermal behavior to be observed if PE was bound to the nanoclay. The peak that corresponded to the maximum degradation in the original dye at 145 °C could not be found or differentiated from the water lost in both nanoclays.

### 3.4. Total Solar Reflectance TSR (%)

The total solar reflectance coefficient TSR (%) calculation subtraction (1—TSR) corresponds to the degree of nanoclay/hybrid pigment's total solar absorption. This calculation requires taking raw reflectance data and applying solar weighting factors for each collected wavelength (ASTM G173–03 [51]). Pigments with high TSR (%) values are designed as cool pigments because of the solar reflectance capacity within the UV-VIS-NIR range. The TSR (%) factor for hybrid pigments could be high due to the intervention of nanoclay particles in solar reflectance, especially in the NIR zone. As we see in Table 5, the TSR values of all the synthesized samples exceeded 50%. However, there were some differences depending on the synthesis conditions. We performed an ANOVA to know if these differences were significant and related to the selected factors. Some factors and interactions

were significantly related to the TSR values for each sample. Figures 13 and 14 represent the significant influences of each factor and interaction.

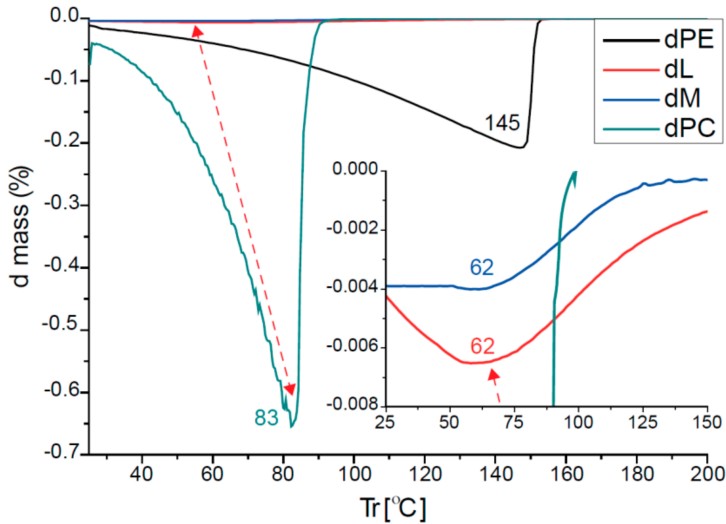

**Figure 10.** DTA diagrams from the original components; montmorillonite (M), laponite (L) phycocyanin (PC) and phycoerythrin (PE).

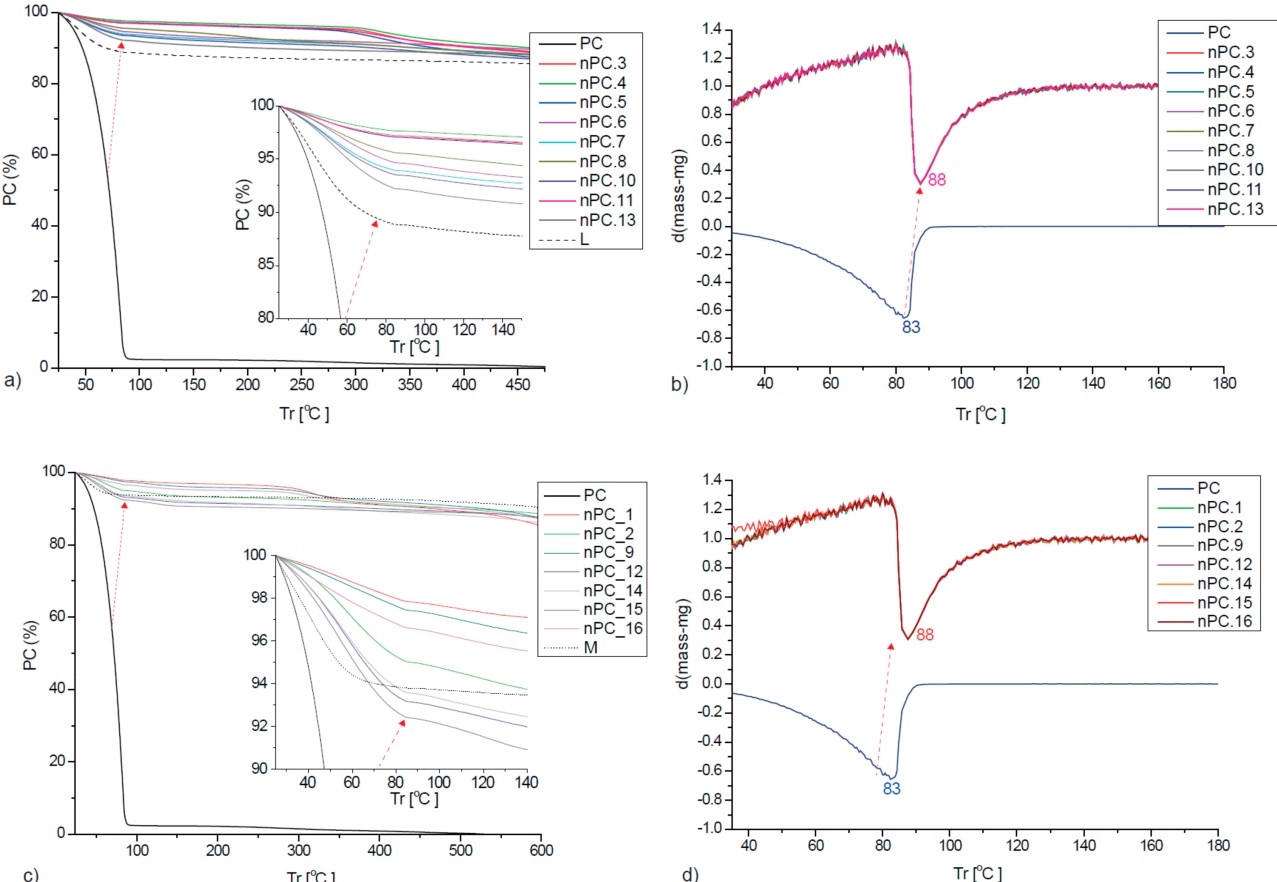

**Figure 11.** (**a**) TGA and (**b**) DTA diagram of laponite (L), PC original pigment and the synthesized hybrid pigments under different conditions [1–16]. (**c**) TGA and (**d**) DTA diagrams of laponite (L), PC original pigment and the synthesized hybrid pigments under different conditions [1–16].

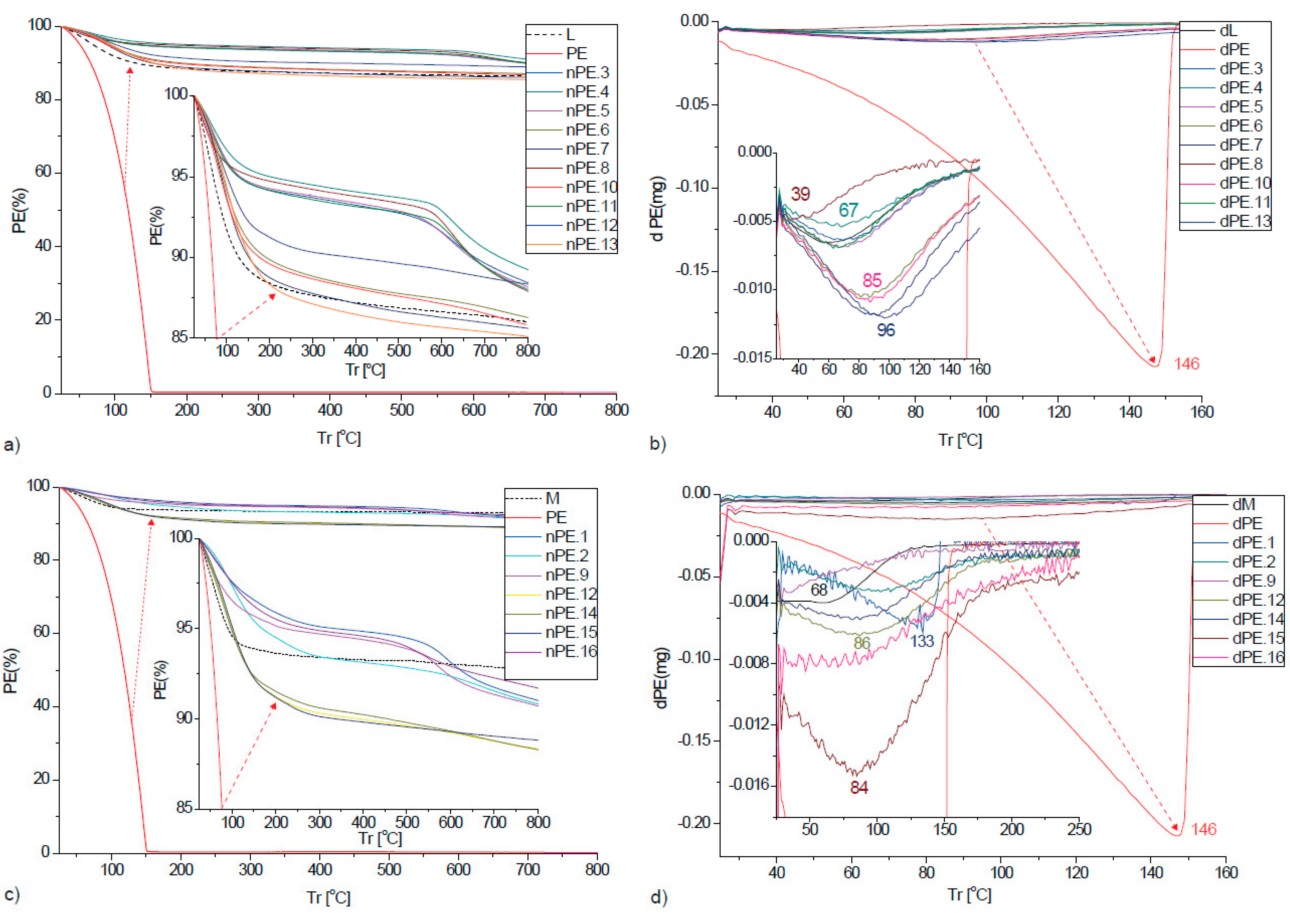

**Figure 12.** (**a**) TGA and (**b**) DTA diagram of laponite (L), PE original pigment and the synthesized hybrid pigments under different conditions [1–16]. (**c**) TGA and (**d**) DTA diagram of laponite (L), PE original pigment and the synthesized hybrid pigments under different conditions [1–16].

**Table 5.** TSR (%) Values from the PC and PE hybrid pigments under different conditions [1–16].

| REF. | TSR (%) | REF. | TSR (%) |
|---|---|---|---|
| nPC.1 | 52.02 | nPE.1 | 54.95 |
| nPC.2 | 39.78 | nPE.2 | 49.24 |
| nPC.3 | 53.58 | nPE.3 | 54.36 |
| nPC.4 | 59.71 | nPE.4 | 58.97 |
| nPC.5 | 48.76 | nPE.5 | 47.76 |
| nPC.6 | 62.84 | nPE.6 | 63.13 |
| nPC.7 | 36.46 | nPE.7 | 48.99 |
| nPC.8 | 41.76 | nPE.8 | 69.10 |
| nPC.9 | 50.20 | nPE.9 | 63.00 |
| nPC.10 | 42.86 | nPE.10 | 55.71 |
| nPC.11 | 46.01 | nPE.11 | 61.15 |
| nPC.12 | 47.31 | nPE.12 | 55.93 |
| nPC.13 | 43.48 | nPE.13 | 50.78 |
| nPC.14 | 40.81 | nPE.14 | 50.89 |
| nPC.15 | 52.95 | nPE.15 | 54.32 |
| nPC.16 | 53.36 | nPE.16 | 54.29 |

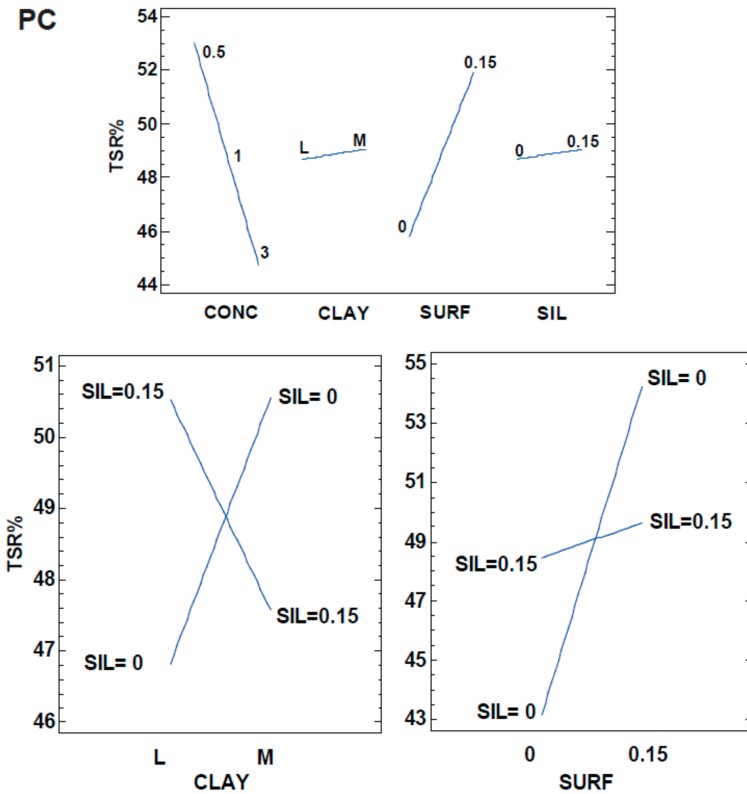

**Figure 13.** Main effects and interactions plots from the TSR (%) response in the PC pigments.

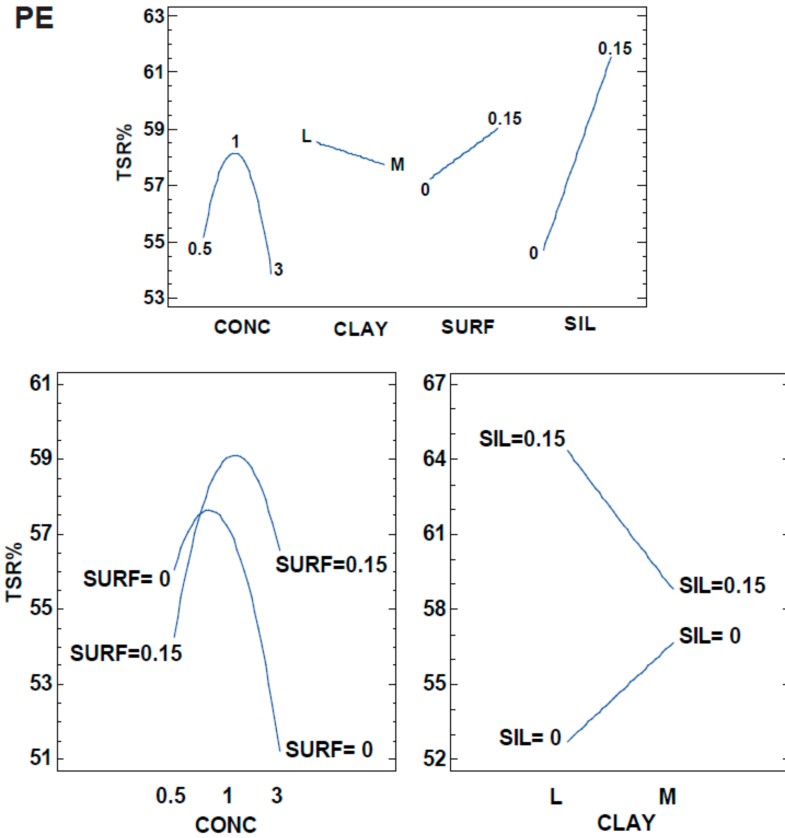

**Figure 14.** Main effects and interactions plots from the TSR (%) response in the PE pigments.



In the PC samples, nanoclay origin was not significant, but the clay and silane concentration interactions were. With L, the best option was to use silane to increase the pigment TSR (%) values, while the opposite was true for M when using silane is not recommended. The dye concentration was also significant and, as expected, the low concentrated pigments obtained high TSR (%) values because of their high lightness values. The surfactant significantly increased the TSR (%) values when not combined with silane.

In the PE pigments, the nanoclay and silane interaction was significant as it was in the PC pigments. Like before, the maximum TSR (%) value was obtained by L when silane was present. With M, no significant differences appeared due to silane interactions. In this pigment, the medium 1% concentration obtained a high TSR (%) value, and the presence of a surfactant increased the TSR (%) value combined with the highest 3% dye content. As previously mentioned, the color perception of these parameters matched the less chromatic and lighter samples, but their TSR (%) values rose.

*3.5. XRD*

XRD patterns show how both nanoclays' crystal structures changed due to PC and PE adsorption. All the hybrid samples with the M nanoclay significantly decreased $2\theta^o$, which corresponded to a basal space (d(001) plane) increment. In the L nanoclay, there were samples in which differences were significant and the interlayer space clearly increased, as well as samples in which the basal space was the same as that we found in the original nanoclay, or even less (Figure 15). In these samples, we could not ensure that the PC or PE interactions occurred into the basal space. We deduced that the dye clay interactions took place on the nanoclay surface, and protection was lower than in the other samples.

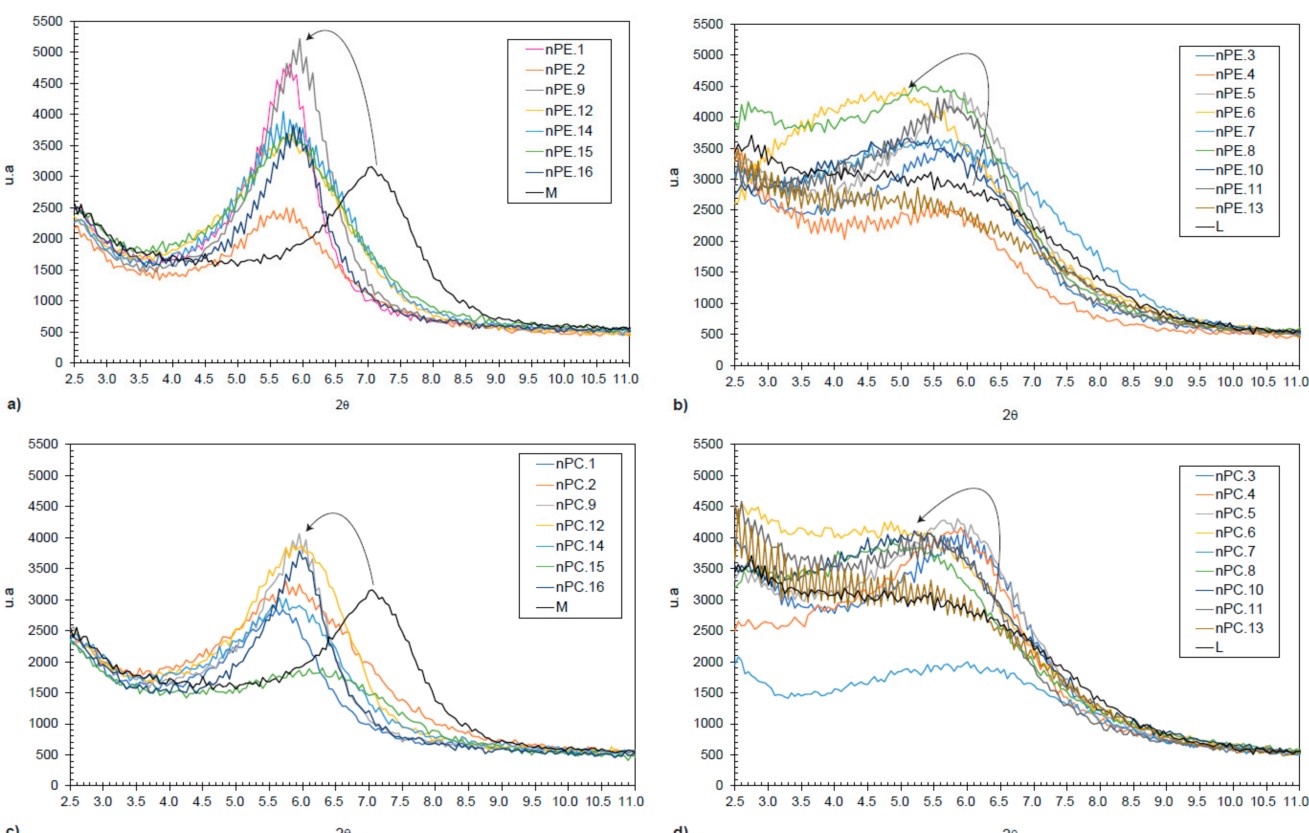

**Figure 15.** XRD patterns from; PE hybrid pigments with M (**a**), L (**b**) clays, PC hybrid pigments with M (**c**), and L (**d**) clays.

The only way to know the effect of the synthesis factors in the interlayer space increment was by the ANOVA analysis. We used interlaminar space $d_{001}$ as a response for the analysis in both hybrid pigments from PC and PE (Table 6). The *p*-value from both

analyses was below 0.05 with the general models, and with some interactions and single factors. The model adjustment in the PC hybrid pigments was lower than the adjustment with the PE pigments. Interactions were all significant in the PE samples, but only the AD interaction was significant in the PC samples (Table 7).

**Table 6.** Interlaminar space d(001)nm from the M and L original clays, and hybrid pigments with PE and PC under different synthesis conditions [1–16].

| PE | 2θ | d(001) nm | PC | 2θ | d(001) nm |
|---|---|---|---|---|---|
| L | 6.10 | 1.45 | L | 6.10 | 1.45 |
| nPE.3 | 6.00 | 1.47 | nPC.3 | 6.00 | 1.47 |
| nPE.4 | 5.80 | 1.52 | nPC.4 | 6.00 | 1.47 |
| nPE.5 | 5.85 | 1.51 | nPC.5 | 6.00 | 1.47 |
| nPE.6 | 4.80 | 1.84 | nPC.6 | 6.00 | 1.47 |
| nPE.7 | 6.00 | 1.47 | nPC.7 | 6.20 | 1.42 |
| nPE.8 | 5.70 | 1.55 | nPC.8 | 5.10 | 1.73 |
| nPE.10 | 5.40 | 1.64 | nPC.10 | 5.40 | 1.64 |
| nPE.11 | 5.90 | 1.50 | nPC.11 | 5.40 | 1.64 |
| nPE.13 | 5.85 | 1.51 | nPC.13 | 5.75 | 1.54 |
| M | 7.05 | 1.25 | M | 7.05 | 1.25 |
| nPE.1 | 5.75 | 1.54 | nPC.1 | 5.60 | 1.58 |
| nPE.2 | 5.70 | 1.55 | nPC.2 | 6.00 | 1.47 |
| nPE.9 | 5.95 | 1.48 | nPC.9 | 5.95 | 1.48 |
| nPE.12 | 5.75 | 1.54 | nPC.12 | 5.95 | 1.48 |
| nPE.14 | 5.80 | 1.52 | nPC.14 | 5.90 | 1.50 |
| nPE.15 | 5.85 | 1.51 | nPC.15 | 6.45 | 1.37 |
| nPE.16 | 5.95 | 1.48 | nPC.16 | 6.00 | 1.47 |

**Table 7.** ANOVA analysis for the interlaminar space d(001)nm response from the hybrid pigments with PC and PE under different synthesis conditions [1–16].

| | | | PC Analysis of Variance | | |
|---|---|---|---|---|---|
| **Source** | **[a] Sum Sq.** | **[b] d.f.** | **[c] Mean Sq.** | **F-Ratio** | **$p$-Value** |
| A:CONC | 0.0106571 | 1 | 0.0106571 | 3.13 | 0.1074 |
| B:CLAY | 0.0195192 | 1 | 0.0195192 | 5.73 | 0.0377 |
| C:SURF | 0.00210412 | 1 | 0.00210412 | 0.62 | 0.4502 |
| D:SIL | 0.0284249 | 1 | 0.0284249 | 8.34 | 0.0162 |
| AD | 0.0239061 | 1 | 0.0239061 | 7.02 | 0.0244 |
| Error total | 0.0340763 | 10 | 0.00340763 | | |
| Total (corr.) | 0.1271 | 15 | | | |
| R2 | 73.1894 | | | | |
| | | | **PE Analysis of Variance** | | |
| A:CONC | 0.00515327 | 1 | 0.00515327 | 8.59 | 0.0326 |
| B:CLAY | 0.000730655 | 1 | 0.000730655 | 1.22 | 0.3201 |
| C:SURF | 0.0111568 | 1 | 0.0111568 | 18.59 | 0.0076 |
| D:SIL | 0.00584537 | 1 | 0.00584537 | 9.74 | 0.0262 |
| AB | 0.0118641 | 1 | 0.0118641 | 19.77 | 0.0067 |
| AC | 0.0249556 | 1 | 0.0249556 | 41.59 | 0.0013 |
| AD | 0.00799175 | 1 | 0.00799175 | 13.32 | 0.0148 |
| BC | 0.0165288 | 1 | 0.0165288 | 27.55 | 0.0033 |
| BD | 0.038032 | 1 | 0.038032 | 63.38 | 0.0005 |
| CD | 0.023214 | 1 | 0.023214 | 38.69 | 0.0016 |
| Error total | 0.00300031 | 5 | 0.000600063 | | |
| Total (corr.) | 0.122294 | 15 | | | |
| R2 | 97.5466 | | | | |

[a] Sum of Squares, [b] Degrees of freedom, [c] Mean of Squares.

For the PC pigments, the only significant factor was nanoclay, and the only significant interaction was concentration with the silane additive. In the L nanoclay, the dye intercalation effect was stronger than the effect on the M clay. At the lowest PC concentration, silane treatment had no effect. At the maximum PC concentration, the silane additive had to be used to increase the basal space from both nanoclays (Figure 16). The surface modification of clays L [52] and M [53] with hydrophobic groups considerably enhanced the intercalation of bulky hydro-phobic molecules in both structures.

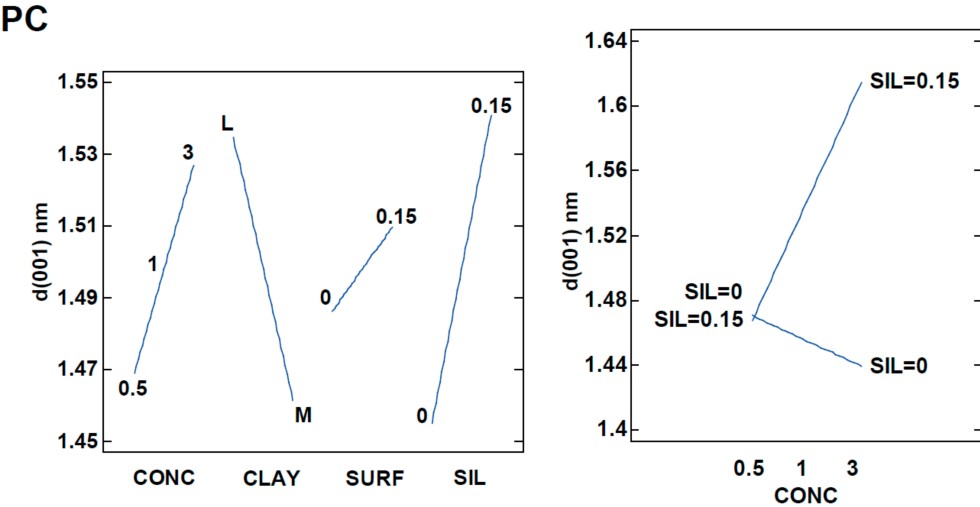

**Figure 16.** Main effects and interactions plots from the d(001)nm response in the PC pigments.

Finally, the results showed the importance of using a surfactant at the maximum PE concentration to increase the dye loaded onto the nanoclay structure, which also occurs for other examples with laminar nanoclays [54]. However, silane was not significant at a high concentration, but was significant at a lower concentration. Under these conditions, silane improved PE adsorption in the nanoclay structures. To achieve the maximum $d_{001}$ distance in M, the best conditions were achieved by using the maximum PE concentration. With L, the maximum separation was found at the minimum PE concentration using the silane additive (Figure 17).

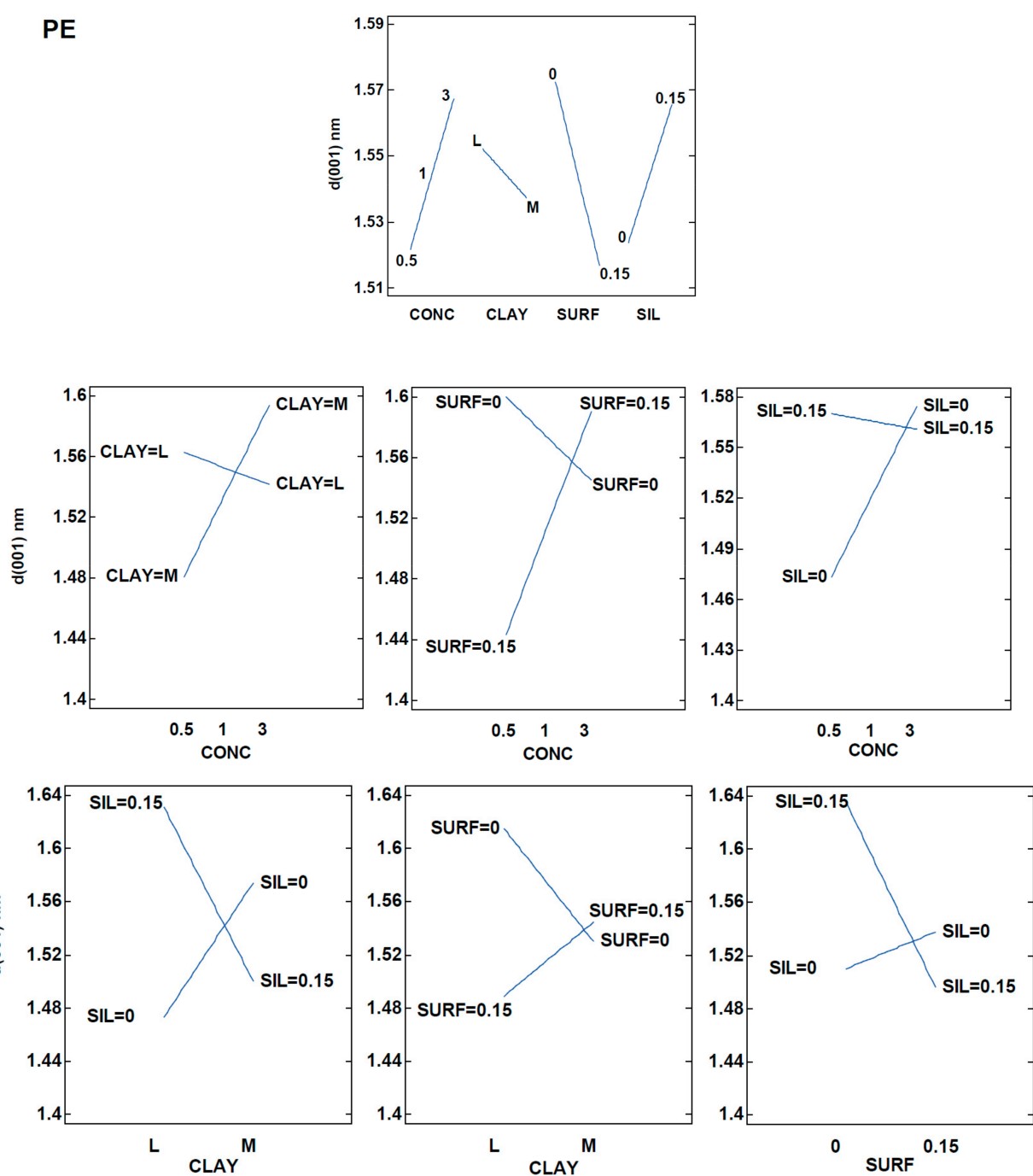

**Figure 17.** Main effects and interactions plots from the d(001) nm response in the PE pigments.

## 4. Conclusions

As we did before with other natural dyes, we successfully combined PC and PE samples in two different laminar nanoclays: natural M and synthetic L. The XRD patterns show how both structures were changed by both PC and PE intercalations. The maximum $d_{001}$ in the PC hybrid pigments was found with the L clay using a silane modifier. With PE the best conditions for both clays appeared at a high dye concentration. With M, it was better to use a surfactant to open the clay structure and load the dye in the basal space. With L, the maximum $d_{001}$ was achieved using the silane additive.

After protein adsorption under different conditions, both nanoclays stabilized PC and PE by allowing their manipulation at room temperature. It was no longer necessary to avoid light at $-4\ °C$ given their initial solutions (buffer phosphate preparations).

Synthesis performance, and optical and thermal properties, depend on the nanoclay structure, the concentration of both dyes added during the synthesis process, and also on additives, the surfactant and/or silane addition, as we explored before with other natural dyes [55]. This time, however, the slight stability of both proteins was extremely difficult, which could be avoided by controlling pH during adsorption.

The maximum adsorption in both dyes at the maximum dye concentration was achieved when silane was employed after dye addition, as we studied before [56]. Nevertheless, the presence of silane decreased dye adsorption in PE, but increased adsorption in PC. We performed all the ANOVAs for each dye separately because of the significant differences found.

The color gamut in both dyes was wide and very interesting. To acquire darker samples as a starting point for a wide color gamut, the high dye concentration must be used with both PC and PE. With PC and L, the most interesting colors were acquired without employing the surfactant and silane additives. The M clay gave more chromatic results at the high concentration than L. In PE, the most chromatic samples were achieved with neither the surfactant nor silane in the L clay. In these samples, less chromatic and lighter samples were found at the medium concentration level (1%) due to the agglomeration phenomenon in the nanoclay structure at this concentration.

Both dyes' intercalations in the nanoclay structures were reflected in the thermal patterns because of the changes in structural water loss. In PC, the lesser thermal resistant dye, thermal stabilization was observed due to nanoclay-dye interactions.

Finally, the TSR (%) values of all the synthesized pigments went up to 50%, which makes all these pigments interesting for coating applications as cool pigments. Moreover, depending on the synthesis conditions, the TSR values significantly affected the results. In relation to dye adsorption and color perception, both additives, silane and the surfactant, inhibited the dye–clay interaction. When more dye was lost, samples were lighter and less chromatic, but the TSR (%) values were high. This is why these factors must be controlled according to the final application.

Agglomeration is a risk that must be taken into account for future applications. For example, if these hybrids are to be used as polymer additives to improve their optical properties. Polymer properties will improve if hybrid pigments are exfoliated in the polymer matrix. Therefore, depending on the application, the silane and surfactant can be important to not only increase the adsorbed PC or PE in nanoclays' structures, but to also improve particle exfoliation in a polymer matrix [57,58]. Other potential applications are packaging, including the food packaging, cosmetics, textile industries, as pigments for stamp or additives for filaments (spinning) and 3D printing (filaments or resin baths) [59–61]. In all the industrial applications, size control is important and can be the reason for select L or M as a nanoclay.

**Supplementary Materials:** The following are available online at https://www.mdpi.com/article/10.3390/app112411992/s1, Table S1: TGA_PE_supplementary 1, Table S2: TGA_PE_supplementary 2.

**Author Contributions:** Writing—Original Draft Preparation, B.M.-V.; Methodology, J.J.-N., B.M.-V. and R.B.; Software, J.J.-N. and B.M.-V.; Validation, E.P.R. and R.B.; Formal Analysis, J.J.-N., B.M.-V. and R.B.; Investigation, J.J.-N. and B.M.-V.; Resources, E.P.R. and V.V.; Data Curation, B.M.-V. and J.J.-N.; Writing—Review and Editing, E.P.R. and R.B.; Supervision, R.B. and V.V.; Project Administration, V.V.; Funding Acquisition, V.V. and J.P. All authors have read and agreed to the published version of the manuscript.

**Funding:** This research was funded by Spanish Ministry of Economy and Competitiveness (Project RTI2018-096000-B-100).

**Institutional Review Board Statement:** Not applicable.

**Informed Consent Statement:** Not applicable.

**Acknowledgments:** To the Spanish Ministry of Economy and Competitiveness for funding the project entitled: "Diseño y caracterización de la apariencia visual de productos" (REF: RTI2018-096000-B-100).

**Conflicts of Interest:** The authors declare no conflict of interest.

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
