# Peer review of "Using Laminar Nanoclays for Phycocyanin and Phycoerythrin Stabilization as New Natural Hybrid Pigments from Microalgae Extraction"

_applsci, doi:10.3390/app112411992_

Round 1

Reviewer 1 Report

The manuscript ,,Using laminar nanoclays for phycocyanin and phycoerythrin stabilization and applications as new natural hybrid pigments from microalgae extraction,, contains many interesting results. However, the manuscript needs to be significantly improved before it is considered for publication.

I propose a major revision.

My recommendations, advice and comments are listed below:

Please be sure that your manuscript thoroughly establishes how this work is fundamentally novel. Specific comparisons should be made to previously published materials that have a similar purpose. Please present a strong case for how this work is a major advance. This needs to be done in the manuscript itself, not just in the response to review comments. 

Please be sure that your abstract and your Conclusions section not only summarize the key findings of your work but also explain the specific ways in which this work fundamentally advances the field relative to prior literature.

The significance of this study should be more emphasize in the introduction.

The abstract should be improved and should be factual. It should contain objectives, results and conclusions.

The language of the manuscript should be improved.

Line 36: I think the handwriting should contain a maximum of 6 keywords so as not to be confusing.

Line 101: Give the structural formula and the size of the charge and CEC of natural montmorillonite.

Line 108: Indicate the purity and molar concentration of all chemicals used for your experiments.

Line 214: I would be interested in standard deviations. Please list them in the table.

Line 245: The same here, too. 

If the figures 5 - 8 could be presented in a better way, I would welcome it.

Line 312: Figure 9 needs to be improved.

Line 316 and 319: It is necessary to denote a, b, c, d etc. in the name. These figures should also be improved.

Line 374: Indicate the possible risks of such research. Add your recommendations for future research.

Line 415: Make sure the references are added correctly according to the journal's instructions.

Author Response

The manuscript, Using laminar nanoclays for phycocyanin and phycoerythrin stabilization and applications as new natural hybrid pigments from microalgae extraction, contains many interesting results. However, the manuscript needs to be significantly improved before it is considered for publication.

I propose a major revision.

My recommendations, advice and comments are listed below:

Please be sure that your manuscript thoroughly establishes how this work is fundamentally novel. Specific comparisons should be made to previously published materials that have a similar purpose. Please present a strong case for how this work is a major advance. This needs to be done in the manuscript itself, not just in the response to review comments. 

The reviewer is right, for this reason we add the highlights of the work in the text (final paragraph of the Introduction section) as you can see here:

Other research have been working with the dye adsorption into the nanoclays structures [32]. We select natural proteins extracted from microalgae biomass (PC and PE) as guest in the nanoclay compounds. We also want to compare the nanoclay structure, silane and surfactant interactions in the PC and PE adsorption [33]. In addition we are going to compare the optical properties of the hybrid pigments synthetized in different conditions in order to know the optimal combination to achieve a wide colour gamut with both proteins.

Please be sure that your abstract and your Conclusions section not only summarize the key findings of your work but also explain the specific ways in which this work fundamentally advances the field relative to prior literature.

The reviewer is right, for this reason we add the following explanation in both sections:

Abstract: C-Phycocyanin (PC) and B-phycoerythrin (PE) are light-harvesting and water-soluble phycobiliproteins from microalgae that belong mainly to the cyanobaceria and rhodhophytes families. Different methods have been developed for PC and PE extraction and purification from microalgae, and offer a high potential for their use as additives in sectors like food, cosmetics, among others. However, the main limitation with using these dyes is their environmental factors sensitivity, such as light fastness, temperature and pH. We employed safe lamellar nanoclays, such as montmorillonite (M) and Laponite (L), for phycobiliproteins stabilization and we do it successfully, as we did it before with other natural dyes. We obtained a wide color gamut from blues to pink by combining four different factors under synthesis conditions: three dye concentrations, two laminar nanoclay sizes, and a two nanoclay surface modifiers combination with cetylpyridinium bromide (CPB) and a coupling agent (3-Aminopropyl) triethoxysilane. The experimental conditions were defined according to a multilevel factorial design of experiment (DOE), in order to study the factors interactions in the final hybrid pigment characteristics. The optical and thermal PE and PC properties significantly improved. We show the optimal synthesis conditions to increase the PC and PE adsorption using the high dye concentration, with surfactant and silane depending on the nanoclay. The hybrid pigments from these phycobiliproteins offer the opportunity to perform several industrial applications as polymer additives, cosmetics, packaging etc.

Conclusions:

The Agglomeration is a risk that must be taken into account for future applications. For example if these hybrids will be used as polymer additives in order to improve their optical properties [58].  The polymer properties will be improves if, the hybrid pigments are exfoliated in the polymer matrix. For these reason depending on the application the silane and surfactant can be important not only to increase the adsorbed PC or PE into the nanoclays structures, can be important to improve the particle exfoliation into a polymer matrix [59]. Other potential applications can be packaging, including food packaging, cosmetics, textile industry as pigments for stamp or additives for filaments (spinning), 3D printing (filament or at the resin bath) etc [60-63]. In all the industrial application the size control will be important and can be the reason to select Laponite or Montmorillonite as the nanoclay.        

The significance of this study should be more emphasize in the introduction.

We do it as we explain in the first comment.

The abstract should be improved and should be factual. It should contain objectives, results and conclusions.

The reviewer is right and we do it as you can see in the new version above.

The language of the manuscript should be improved.

We have sent the manuscript for an English revision by native English. In addition, due to the reviewers suggestions we have redone it again, including our new improvements to a new revision.

Line 36: I think the handwriting should contain a maximum of 6 keywords so as not to be confusing.

Following the reviewer suggestions we decided to reduce the keywords in: Keywords: phycocyanin; phycoerythrin; stabilization; statistical design of experiments; Montmorillonite; Laponite; hybrid pigments.

Line 101: Give the structural formula and the size of the charge and CEC of natural montmorillonite.

Following the reviewer comments, we add the following information in 2.1 section: The chemical composition for montmorillonite is 80% SiO2, 13% Al2O3, and 3% Fe2O3 (molecular formula: Al2H2O12Si4) [39]. The montmorillonite cation exchange capacity (CEC), is approximately 90–145 cmol·kg−1. Synthetic Lap (Si8[Mg5.5Li0.4H4.0O24]0.7Na0.7+) is a 2D clay disc-shaped silicate that is approximately 1 nm thick. Its diameter is 25 nm. The adsorption capacity of laponite-based hydrogels and the CEC is 74 cmol·kg−1[40].

  1. Chong, A.S.; Manan, M.A.; Idris, A.K. Readiness of Lignosulfonate Adsorption onto Montmorillonite. Colloids Surfaces A Physicochem. Eng. Asp. 2021, 628, 127318, doi:10.1016/j.colsurfa.2021.127318.
  2. Wang, Z.; Li, J.; Sun, Y.; Peng, J.; Wang, J.; Hao, Y.; Li, W.; Zhang, P.; Ning, W.; Miao, S. Laponite Elementary Sheets Assisted Fluorescence Resonance Energy Transfer: A Demonstration by Langmuir-Blodgett Technique. Dye. Pigment. 2021, 196, 109800, doi:10.1016/j.dyepig.2021.109800.

Line 108: Indicate the purity and molar concentration of all chemicals used for your experiments. Following the reviewer comments, we add the information as you can see:

They were supplied by Southern Clay Products (Gonzales, TX, USA) and Rockwood, respectively. As nanoclay organic modifiers, we employed surfactant cetylpyridinium bromide (SURF) C21H38BrN·6H2O, 384.44 g/mol, purity 98%, and silane compound (SIL) 3-aminopropyltriethoxysilane, H2N(CH2)3Si(OCH3)3, 179.29 g/mol, purity 99%. pH variation during the synthesis process was controlled by chloridric acid HCl- (37%) all supplied by Sigma-Aldrich.

Line 214: I would be interested in standard deviations. Please list them. Following the suggestion, we add the standard deviations in the table caption for each kind of samples (Phycoerythrin and Phycocyanin). The deviations are high due to the synthesis factor that we selected in the DOE. However, in Table 3 we can se the error calculation by groups and between the groups.

a PE sample standard deviation: 27.2457

b PC sample standard deviation: 8.7568

Table 3. Analysis of variance from PC and PE adsorptions (%) under the DOE conditions.

PC

Source

aSum Sq.

bd.f.

cMean Sq.

F-Ratio

P-Value

A:CONC

208.407

1

208.407

14.85

0.0063

B:CLAY

88.1283

1

88.1283

6.28

0.0406

C:SURF

0.153236

1

0.153236

0.01

0.9197

D:SIL

249.591

1

249.591

17.78

0.004

AB

9.88922

1

9.88922

0.7

0.429

AD

166.704

1

166.704

11.88

0.0107

BD

113.902

1

113.902

8.12

0.0247

CD

13.9593

1

13.9593

0.99

0.3518

Error total

98.2455

7

14.0351

Total (corr.)

1150.21

15

R2: 78.20%

PE

Source

aSum Sq.

bd.f.

cMean Sq.

F-Ratio

P-Value

A:CONC

8355.89

1

8355.89

66.75

0.0001

B:CLAY

392.87

1

392.87

3.14

0.1198

C:SURF

222.976

1

222.976

1.78

0.2238

D:SIL

70.5179

1

70.5179

0.56

0.4774

AA

509.76

1

509.76

4.07

0.0834

AB

395.824

1

395.824

3.16

0.1186

AC

935.866

1

935.866

7.48

0.0292

BD

450.666

1

450.666

3.6

0.0996

Error total

876.325

7

125.189

Total (corr.)

11134.9

15

R2: 90.44%

a Sum of Squares

b Degrees of freedom

c Mean of Squares

Line 245: The same here, too. As we did the ANOVA models for the Adsorption parameters, and we indicate the sum of squares and the medium error we add the R square in the tables to see the multivariate adjustment in both pigments.

If the figures 5 - 8 could be presented in a better way, I would welcome it.

We improved the figure presentations including legend to make easy their interpretation as you can see in the example:

Line 312: Figure 9 needs to be improved.

We did it, improving the typography, the kerning and including arrows to enable understanding  the main information.

Line 316 and 319: It is necessary to denote a, b, c, d etc. in the name. These figures should also be improved.

We do it as it can be seen in the new figure version. We improved again the kerning in the legend, increase the lines size and add the a,b,c,d legends to each one.

Figure 10. a) TGA and b) DTA diagram of laponite (L), PC original pigment and the synthesized hybrid pigments under different conditions [1-16]. c) TGA and d) DTA diagram of laponite (L), PC original pigment and the synthesized hybrid pigments under different conditions [1-16].

Line 374: Indicate the possible risks of such research. Add your recommendations for future research.

Following the reviewer suggestions we add the paragraph in the conclusion section: The Agglomeration is a risk that must be taken into account for future applications. For example if these hybrids will be used as polymer additives in order to improve their optical properties [58]. The polymer properties will be improved if, the hybrid pigments are exfoliated in the polymer matrix.

Line 415: Make sure the references are added correctly according to the journal's instructions.

We did it and Mendeley was used to follow the journal style.

Reviewer 2 Report

 The article is a well written one; also it is rich in information, However the group has already published a number of articles in the area (not only References 31 and 37). What the group has published in this area along with the contrubutions of others make the actual value of the pertinent new articles of the present group a diminishing one; as a result a similar potential article byt the same group might not deserve publication and it is recommended that the group will reorient substantially their further research if the expect publication in jounal having some substantial impact. As regards the present article it is recommended that the authors will review in more detail ALL their pertinent wok in the introduction and also, at the end, compare clearly their new findings with what is known already (from previous work by them and others).

Author Response

REVIEWER 2

The article is a well written one; also it is rich in information, However the group has already published a number of articles in the area (not only References 31 and 37). What the group has published in this area along with the contrubutions of others make the actual value of the pertinent new articles of the present group a diminishing one; as a result a similar potential article by the same group might not deserve publication and it is recommended that the group will reorient substantially their further research if the expect publication in journal having some substantial impact. As regards the present article it is recommended that the authors will review in more detail ALL their pertinent wok in the introduction and also, at the end, compare clearly their new findings with what is known already (from previous work by them and others).

The reviewer is right we would try to avoid to abuse with the self-citation. However, following the suggestions we increased the comments and citations from our previous works in order to indicate the contribution of this work and future applications due to these new findings as you can see for example: 

When hybrid nanopigments are exfoliated into a polymer matrix, the optical, thermal and mechanical properties of that matrix improves, which provides possibilities for original dye applications [31].  As green chemistry trend, recent studies have been work on the stabilization of natural dyes, or synthetic similar synthetic cations, onto inorganic substrates which results in the formation of hybrid pigments. For example the used of flavylium cations as synthetic analogues to anthocyanins demonstrated that laminar nanoclays as montmorillonite and laponite, can adsorb efficiently this cations to create hybrid pigments with improved properties [32]. To overcome the poor stability of natural lutein to environmental factors, hydrotalcite was incorporated by a green mechanical grinding process. The thermal decomposition of lutein/LDH was significantly improved by the chemical interaction achieved [33]. Vermiculite improves hybrids pigment using alizarin cations improves elastomer flam-mability properties [34]. Aluminum-magnesium hydroxycarbonate (LH) modified with carmic acid improves Polymmer composites properties reducing the polymer matrix flammability. In addition, the stabilization of CA on LH leads to pigments with an excellent resistance to acetone and water[35]. In conclusion clay minerals are promising carriers for enhancing the stability of plant pigments. It is possible an optimal and promising strategy to enhance the stability of nat-ural pigments against various chemical reagents, high temperature and visible light irra-diation after incorporation of natural or synthetic clay minerals[36].

The surface modifiers concentration fell within a 0-1% range over nanoclay mass. The solvent separation and drying process by lyophilization were carried on as in our previous works [38].

The synthesis performance, and optical and thermal properties, depend on the nanoclay structure, the concentration of both dyes added during the synthesis process, and also on additives, the surfactant and/or silane addition, as we explore before with other natural dyes[9]. However the low stability from both proteins was an extra difficult this time, that we could avoid controlling the pH during the adsorption.

  1. Micó-Vicent, B.; Ramos, M.; Viqueira, V.; Luzi, F.; Dominici, F.; Terenzi, A.; Maron, E.; Hamzaoui, M.; Kohnen, S.; Torre, L.; et al. Anthocyanin Hybrid Nanopigments from Pomegranate Waste: Colour, Thermomechanical Stability and Environmental Impact of Polyester-Based Bionanocomposites. Polymers 2021, 13.
  2. Micó-Vicent, B.; Jordán, J.; Martínez-Verdú, F.; Balart, R. A Combination of Three Surface Modifiers for the Optimal Generation and Application of Natural Hybrid Nanopigments in a Biodegradable Resin. Journal of Materials Science 2017, 52, doi:10.1007/s10853-016-0384-8.
  3. Micó-Vicent, B.; Jordán, J.; Perales, E.; Martínez-Verdú, F.M.; Cases, F.; Micó-Vicent, B.; Jordán, J.; Perales, E.; Martínez-Verdú, F.M.; Cases, F. Finding the Additives Incorporation Moment in Hybrid Natural Pigments Synthesis to Improve Bioresin Properties. Coatings 2019, Vol. 9, Page 34 2019, 9, 34, doi:10.3390/COATINGS9010034.
  4. Micó-Vicent, B.; Perales Romero, E.; Jordán-Núñez, J.; Viqueira, V. Halloysite and Laponite Hybrid Pigments Synthesis with Copper Chlorophyll. Applied Sciences 2021, 11.
  5. Micó-Vicent, B.; Ramos, M.; Luzi, F.; Dominici, F.; Viqueira, V.; Torre, L.; Jiménez, A.; Puglia, D.; Garrigós, M.C. Effect of Chlorophyll Hybrid Nanopigments from Broccoli Waste on Thermomechanical and Colour Behaviour of Polyester-Based Bionanocomposites. Polymers 2020, 12, 2508.
  6. Micó-Vicent, B.; Jordán, J.; Perales, E.; Martínez-Verdú, F.; Cases, F. Finding the Additives Incorporation Moment in Hybrid Natural Pigments Synthesis to Improve Bioresin Properties. Coatings 2019, 9.

Reviewer 3 Report

In the present work, the authors showed their studies on the preparation of hybrid pigments based on phycocyanin and phycoerythrin colorants and two laminar nanoclays. The prepared hybrid pigments have been analyzed in terms of their optical and thermal properties. The stabilization of natural colorants and the designing of new functional hybrid systems is becoming a popular and important issue nowadays. Therefore, the topic addressed by the authors is relevant and interesting and fits the scope of the Applied Sciences journal. However, the Reviewer strongly recommends improving the whole manuscript in terms of the discussion and language editing. The scientific novelty of the work in relation to the other studies should also be more emphasized. In my opinion, the abovementioned work can be considered for publication after major corrections:

1) The article title contains information on the application of hybrid pigments, while the study did not show any application research. Could you explain it?

2) What kind of bioresin was employed for this study? How were the pigments introduced into bioresin matrix and what were the properties of these composites? There are no research results related to the application of hybrid pigments in bioresin.

3) Please correct the form of citation (e.g. [4-9] instead of [4],[5],[6],[7],[8],[9]).

4) The scientific novelty should be more emphasized.

5) I recommend improving the text for the English language (there are many mistakes – for example: „However, the application for this pigment…..” – should be: „…of this pigment”)

6) In the introduction part, there is a lack of information on recent reports on the stabilization of other natural dyes onto inorganic substrates which results in the formation of hybrid pigments with improved performance, such as anthyocyanins (see for example: Silva GTM. Et al., ACS omega 2020, 5(41), 26592-26600), lutein (see for example: Li S. et al., Molecules 2020, 25(5), 1231), alizarin (see for example: Marzec A. et al., Dyes and Pigments 2021, 186, 108965), carminic acid (see for example: Marzec A. et al., Molecules 2019, 24(3), 560) and others.

7) As organic components in the hybrid pigments two different colorants have been used – phycoerythrin and phycocyanin. Perhaps, it would be useful for readers to add chemical structures of these compounds in the materials part.

8) I recommend showing figure 1 in the form of a scheme with the particular stages of modification, instead of pictures placed in a chaotic manner.

9) In the methodological section, the authors stated that they used XRD analysis in the work (lines 171-175, page 5). However, in the results part I cannot see the XRD curves and discussion on the crystal structure or the interplanar distances of the studied materials. Please comment.

10) It is well known that hybrid pigments are characterized by outstanding resistance to external factors, such as elevated temperatures or chemical agents. Could the authors show some results about the chemical resistance of the studied hybrid pigments?

11) Figure 2: Why do the values repeat on the ordinate axis (e.g. 0.1 or 0.0)?

12) I recommend changing the subtitle 3.2. from CIELAB for e.g. Color characterization

13) To increase the readability of the results obtained from TGA, a table containing the values of T05%, T10%, T20%, and char residue should be prepared.

14) Page 13, lines 307-310: the authors stated that „PC and PE had different thermal degradation peaks, while PE started degradation at 145ºC and PC at 83ºC, which means that their temperature stability are remarkable. In my opinion, this sentence is confusing, as degradation temperature below 100 °C cannot be ascribed as remarkable because it limits significantly the applicative potential of such colorants. Please comment.

15) In the conclusions, the authors stated that both dyes were intercalated into the nanoclay structures (line 393, page 18). In my opinion, the intercalation of the dye molecules between the layers of nanoclays should be confirmed using XRD analysis.

I strongly want to encourage the authors to overwork the paper as it is providing interesting data for future hybrid materials development, despite the huge number of critical comments.

Author Response

REVIEWER 3

In the present work, the authors showed their studies on the preparation of hybrid pigments based on phycocyanin and phycoerythrin colorants and two laminar nanoclays. The prepared hybrid pigments have been analyzed in terms of their optical and thermal properties. The stabilization of natural colorants and the designing of new functional hybrid systems is becoming a popular and important issue nowadays. Therefore, the topic addressed by the authors is relevant and interesting and fits the scope of the Applied Sciences journal. However, the Reviewer strongly recommends improving the whole manuscript in terms of the discussion and language editing. The scientific novelty of the work in relation to the other studies should also be more emphasized. In my opinion, the abovementioned work can be considered for publication after major corrections:

  • The article title contains information on the application of hybrid pigments, while the study did not show any application research. Could you explain it?

The reviewer is right, we have changed the title in order to avoid the confusion and focus in the main goal for this work: “Using laminar nanoclays for phycocyanin and phycoerythrin stabilization as new natural hybrid pigments from microalgae extraction.”

We can explain why we decided to focus the study on the synthesis and not in the application. In this case the goal was the natural proteins stabilization to increase their applications. Taken into account the weak stability for both dyes like the buffer phosphate preparation and the necessity to keep both at -4ºC, we focus the goal of our research in the possibility of both dyes stabilization using laminar nanoclays. In fact we first try with hydrotalcite like in other works but it was not possible due to the pH during the synthesis process. The high pH value that was not possible to decrease destroys both proteins during the adsorption and no color was observed at the end. We add this information explaining better the highlights in the paper to let clear the final contribution as it can be seen at:

First we tried to use also hydrotalcite nanoclay as we did before with other natural dyeshowever  [41] it was not possible due to the nanoclay pH in dispersion, not even whit the calcinated one. The pH keeps above to 4 and it was not possible to decrease it with HCl- as we did with the montmorillonite or the laponite. Both PC and PE lose their color in that pH conditions and only remain the hydrotalcite optical properties. For this reason, we finally selected montmorillonite and laponite clays.  

  1. Micó-Vicent, B.; Viqueira, V.; Ramos, M.; Luzi, F.; Dominici, F.; Torre, L.; Jiménez, A.; Puglia, D.; Garrigós, M.C. Effect of Lemon Waste Natural Dye and Essential Oil Loaded into Laminar Nanoclays on Thermomechanical and Color Properties of Polyester Based Bionanocomposites. Polymers 2020, 12, 1451.
  • What kind of bioresin was employed for this study? How were the pigments introduced into bioresin matrix and what were the properties of these composites? There are no research results related to the application of hybrid pigments in bioresin.

The reviewer is right, it was our mistake. We included the references in which we used an epoxy bioresin as reference, but in this work we focus on the hybrid pigment properties and the PC an PE stabilization as we explained in the first comment. We correct the mistake and add the corrected explanation in the new version.

Abstract: C-Phycocyanin (PC) and B-phycoerythrin (PE) are light-harvesting and water-soluble phycobiliproteins from microalgae that belong mainly to the cyanobaceria and rhodhophytes families. Different methods have been developed for PC and PE extraction and purification from microalgae, and offer a high potential for their use as additives in sectors like food, cosmetics, among others. However, the main limitation with using these dyes is their environmental factors sensitivity, such as light fastness, temperature and pH. We employed safe lamellar nanoclays, such as montmorillonite (M) and Laponite (L), for phycobiliproteins stabilization and we do it successfully, as we did it before with other natural dyes. We obtained a wide color gamut from blues to pink by combining four different factors under synthesis conditions: three dye concentrations, two laminar nanoclay sizes, and a two nanoclay surface modifiers combination with cetylpyridinium bromide (CPB) and a coupling agent (3-Aminopropyl) triethoxysilane. The experimental conditions were defined according to a multilevel factorial design of experiment (DOE), in order to study the factors interactions in the final hybrid pigment characteristics. In both M and L the d001 distance (nm) was increased due to the PC and PE adsorption. The best conditions to increase the basal space depend on the nanoclay structure, and for M is better to used surfactant while with laponite is the silane modification. In addition, optical and thermal PE and PC properties significantly improved. We show the optimal synthesis conditions to increase the PC and PE adsorption using the high dye concentration, with surfactant and silane depending on the nanoclay. The hybrid pigments from these phycobiliproteins offer the opportunity to perform several industrial applications as polymer additives, cosmetics, packaging etc.

  1. Introduction

Several research still working with the natural dye adsorption optimization by nanoclays structures [37].  We select the natural proteins extracted from microalgae adsorption (PC and PE) as gest in the nanoclay compounds. We also want to compare the nanoclay structure, silane and surfactant interactions in the PC and PE adsorption[38]. In addition we are going to com-pare the optical properties of the hybrid pigments synthetized in different condition in or-der to know the optimal combination to achieve a wide colour gamut with both proteins. In this work, we employed natural M and synthetic Laponite nanoclays for PC and PE stabilization. We used statistical design of experiment (DOE) to combine different factors during the synthesis process to find the optimal conditions for maximum dye adsorption. We aimed to study the factors influencing hybrid pigment properties, such as color gamut, thermal fastness or total solar reflectance (%). The studied responses will be the main benefit of the hybrid pigments generated from microalga sources.

Micó-Vicent, B.; Jordán, J.; Martínez-Verdú, F.; Balart, R. A Combination of Three Surface Modifiers for the Optimal Generation and Application of Natural Hybrid Nanopigments in a Biodegradable Resin. Journal of Materials Science 2017, 52, doi:10.1007/s10853-016-0384-8.

Micó-Vicent, B.; Jordán, J.; Perales, E.; Martínez-Verdú, F.M.; Cases, F.; Micó-Vicent, B.; Jordán, J. Finding the Additives Incorporation Moment in Hybrid Natural Pigments Synthesis to Improve Bioresin Properties. Coatings 2019, Vol. 9, Page 34 2019, 9, 34, doi:10.3390/COATINGS9010034.

  1. Conclusions

As we did before with other natural dyes, we successfully combined PC and PE samples in two different laminar nanoclays: natural MMT and synthetic LAP. XRD patterns show how both structures were changed by both PC and PE intercalations. The maximum d001 in the PC hybrids pigments was found with L clay using silane modifier. With PE the best conditions for both clays were found at high dye concentration. With M is better to used surfactant to open the clay structure an load the dye in the basal space. On the other hand with L the maximum d001 is achieved using the silane additive.

After the proteins adsorption at the different conditions, both nanoclays stabilized PC and PE by allowing their manipulation at room temperature. It was no longer necessary to avoid light with -4ºC, as their initial solutions (buffer phosphate preparations).

The synthesis performance, and optical and thermal properties, depend on the nanoclay structure, the concentration of both dyes added during the synthesis process, and also on additives, the surfactant and/or silane addition, as we explore before with other natural dyes [56]. However the low stability from both proteins was an extra difficult this time, that we could avoid controlling the pH during the adsorption.

The maximum adsorption in both dyes at the maximum dye concentration was achieved when silane was employed after dye addition as we studied before [57]. However, the presence of silane decreased the dye adsorption in PE, but increased adsorption in PC. We performed all the ANOVAs for each dye separately because of the significant differences we found.

The color gamut in both dyes was high and very interesting. To acquire darker samples as a starting point for a wide color gamut, the high dye concentration must be used with both PC and PE. With PC and LAP, the most interesting colors were acquired without employing the surfactant and silane additives. MMT clay gave results that were more chromatic at the high concentration than LAP. In PE, the most chromatic samples were achieved with neither the surfactant nor silane in LAP clay. In these samples, less chromatic and lighter samples were found at the medium concentration level (1%) due to the agglomeration phenomenon in the nanoclay structure at this concentration.

Both dyes’ intercalations in the nanoclay structures were reflected in the thermal patterns because of the changes in the structural water loss. In PC, the lesser thermal resistant dye, thermal stabilization was observed due to the nanoclay-dye interactions.

Finally, the TSR(%) values for all the synthesized pigments went up to 50%, which makes all these pigments interesting for coating applications as cool pigments. Moreover, depending on the synthesis conditions, the TSR values significantly affected the results. In relation to dye adsorption and color perception, both additives, silane and the surfactant inhibited the dye-clay interaction. When more dye was lost, samples were lighter and less chromatic, but the TSR(%) values were high. For these reason, these factors must be controlled according to the final application.

The Agglomeration is a risk that must be taken into account for future applications. For example if these hybrids will be used as polymer additives in order to improve their optical properties [58]. The polymer properties will be improved if, the hybrid pigments are exfoliated in the polymer matrix. For these reason depending on the application the silane and surfactant can be important not only to increase the adsorbed PC or PE into the nanoclays structures, can be important to improve the particle exfoliation into a polymer matrix [59]. Other potential applications can be packaging, including food packaging, cosmetics, textile industry as pigments for stamp or additives for filaments (spinning), 3D printing (filament or at the resin bath) etc [60-63]. In all the industrial application the size control will be important and can be the reason to select Laponite or Montmorillonite as the nanoclay.

  • Please correct the form of citation (e.g. [4-9] instead of [4],[5],[6],[7],[8],[9]).

It was a problem with the reference editor and we have corrected it, thanks.

4) The scientific novelty should be more emphasized. The reviewer is right and we have done it as you can find in the new conclusions and references listed before.

5) I recommend improving the text for the English language (there are many mistakes – for example: „However, the application for this pigment…..” – should be: „…of this pigment”)

The reviewer is right and we have sent the paper for the revision before the new submission as you can see with the software (word) change tracking.  

6) In the introduction part, there is a lack of information on recent reports on the stabilization of other natural dyes onto inorganic substrates which results in the formation of hybrid pigments with improved performance, such as anthyocyanins (see for example: Silva GTM. Et al., ACS omega 2020, 5(41), 26592-26600), lutein (see for example: Li S. et al., Molecules 2020, 25(5), 1231), alizarin (see for example: Marzec A. et al., Dyes and Pigments 2021, 186, 108965), carminic acid (see for example: Marzec A. et al., Molecules 2019, 24(3), 560) and others.

We improved the introduction section adding the information that the reviewer suggested and also we include others like the interesting review: Li, Shue, et al. "Recent researches on natural pigments stabilized by clay minerals: A review." Dyes and Pigments (2021): 109322. As you can see in the new paragraph:

As green chemistry trend, recent studies have been work on the stabilization of natural dyes, or synthetic similar synthetic cations, onto inorganic substrates which results in the formation of hybrid pigments. For example the used of flavylium cations as synthetic analogues to anthocyanins demonstrated that laminar nanoclays as montmorillonite and laponite, can adsorb efficiently this cations to create hybrid pigments with improved properties [32]. To overcome the poor stability of natural lutein to environmental factors, hydrotalcite was incorporated by a green mechanical grinding process. The thermal decomposition of lutein/LDH was significantly improved by the chemical interaction achieved [33]. Vermiculite improves hybrids pigment using alizarin cations improves elastomer flammability properties [34]. Aluminum-magnesium hydroxycarbonate (LH) modified with carmic acid improves Polymmer composites properties reducing the polymer matrix flammability. In addition, the stabilization of CA on LH leads to pigments with an excellent resistance to acetone and water [35]. In conclusion clay minerals are promising carriers for enhancing the stability of plant pigments. It is possible an optimal and promising strategy to enhance the stability of natural pigments against various chemical reagents, high temperature and visible light irradiation after incorporation of natural or synthetic clay minerals [36].

7) As organic components in the hybrid pigments two different colorants have been used – phycoerythrin and phycocyanin. Perhaps, it would be useful for readers to add chemical structures of these compounds in the materials part.

The reviewer is right and we add it in the materials section as the new Figure 1:

The structure of the biliproteins depends on the environment in which they are found (physical state, pH, ionic strength, etc.) and can be a complex mixture of trimers, hexamers or monomers. The Figure 1 shows as an example the three-dimensional structure and their dimensions for phycocyanin and phycoerythrin in hexameric state.

Figure 1. Three dimensional structure for biliproteins used in this work: C-Phycocyanin and B-phycoerythrin in hexameric state. Dimensions are shown in Angstroms.

8) I recommend showing figure 1 in the form of a scheme with the particular stages of modification, instead of pictures placed in a chaotic manner.

We follow the reviewer suggestions and changed the new figure 2 as you can see:

9) In the methodological section, the authors stated that they used XRD analysis in the work (lines 171-175, page 5). However, in the results part I cannot see the XRD curves and discussion on the crystal structure or the interplanar distances of the studied materials. Please comment.

The reviewer is right, we have added the section XRD and put all the results and the discussion. Also we add comments at the conclusions and abstract sections. For example:

XRD patterns show how it changes both nanoclays crystal structures due to the PC and PE adsorption. All hybrid samples with montmorillonite nanoclay decrease significantly the 2qo that corresponds to a basal space (d(001) plane) increment. In L nanoclay there are samples in which the differences are significant and the interlayer space is clearly increased, and samples in which the basal space is the same that we found in the original nanoclay or even less. In those samples is not possible to ensure that the PC or PE interactions occurs into the basal space.

Figure 15. XRD patterns from; PE hybrid pigments with M(a), L (b) clays, PC hybrid pigments with M (c), and L (d) clays.

10) It is well known that hybrid pigments are characterized by outstanding resistance to external factors, such as elevated temperatures or chemical agents. Could the authors show some results about the chemical resistance of the studied hybrid pigments?

The original solutions extracted PC and PE dyes are unstable when attacked by acidic (pH values below 4) and basic (pH values above 9) media. In addition, they are also unstable to heat treatments resulting in temperatures above 50º. This instability is because of the fact that both chemical agents and temperature can cause the denaturation of these protein pigments. However, the thermal analysis show that both proteins increase their thermal resistances as you can see at the section: 3.3. Thermogravimetrical analysis DTA.

11) Figure 2: Why do the values repeat on the ordinate axis (e.g. 0.1 or 0.0)?

The reviewer is right, the ordinate axis were wrong and we correct it as you can find at the next Figure 3:

12) I recommend changing the subtitle 3.2. from CIELAB for e.g. Color characterization.

Perfect, we do it as the reviewer suggested.

13) To increase the readability of the results obtained from TGA, a table containing the values of T05%, T10%, T20%, and char residue should be prepared.

We tried to build it, however the results are more difficult to read and understand in that table. For example, the temperature for the 5% of mass lost for the PE is 38.37ºC, for the M is 58.5ºC, for the L is 70ºC, and for the first hybrid pigment nPE.1 is 181.204ºC. The water substitution in both nanoclays by the dye intercalation decreased the first mass lost in both nanoclays. Is not possible to know the temperature for the 5% of lost in PC or PE hybrid pigments due to the biliprotein degradation because we had a hybrid. In the tables we used the first derivate to know the first peaks of degradation for all the materials in order to make easy the comparisons. The dye concentration in the hybrid pigments is low compared than the mass in the pure dye. For this reason it is difficult to put the results in a table and make comparison. However we add supplementary material with all the results in order to support the discussion of this section.

14) Page 13, lines 307-310: the authors stated that „PC and PE had different thermal degradation peaks, while PE started degradation at 145ºC and PC at 83ºC, which means that their temperature stability are remarkable. In my opinion, this sentence is confusing, as degradation temperature below 100 °C cannot be ascribed as remarkable because it limits significantly the applicative potential of such colorants. Please comment.

Regarding the previous paragraph, we wanted to express that the thermal stability obtained in this article for hybrid pigments (solid state) is remarkable when compared to the stability for pigments in liquid media, where the temperature above which they are already unstable is 55°C for both proteins.

15) In the conclusions, the authors stated that both dyes were intercalated into the nanoclay structures (line 393, page 18). In my opinion, the intercalation of the dye molecules between the layers of nanoclays should be confirmed using XRD analysis.

The reviewer is right, for that we have added the new section XRD and confirm that conclusion and add the information in the concussion discussion and in the abstract as you can see:

C-Phycocyanin (PC) and B-phycoerythrin (PE) are light-harvesting and water-soluble phycobiliproteins from microalgae that belong mainly to the cyanobaceria and rhodhophytes families. Different methods have been developed for PC and PE extraction and purification from microalgae, and offer a high potential for their use as additives in sectors like food, cosmetics, among others. However, the main limitations with using these dyes is their environmental factors sensitivity, such as light fastness, temperature and pH. We employed safe lamellar nanoclays, such as montmorillonite (M) and Laponite (L), for phycobiliproteins stabilization and we do it successfully, as we did it before with other natural dyes. We obtained a wide color gamut from blues to pink by combining four different factors under synthesis conditions: three dye concentrations, two laminar nanoclay sizes, and a two nanoclay surface modifiers combination with cetylpyridinium bromide (CPB) and a coupling agent (3-Aminopropyl) triethoxysilane. The experimental conditions were defined according to a multilevel factorial design of experiment (DOE), in order to study the factors interactions in the final hybrid pigment characteristics. In both M and L the d001 distance (nm) was increased due to the PC and PE adsorption. The best conditions to increase the basal space depend on the nanoclay structure, and for M is better to used surfactant while with laponite is the silane modification. In addition, optical and thermal PE and PC properties significantly improved. We show the optimal synthesis conditions to increase the PC and PE adsorption using the high dye concentration, with surfactant and silane depending on the nanoclay. The hybrid pigments from these phycobiliproteins offer the opportunity to perform several industrial applications as polymer additives, cosmetics, packaging etc.

I strongly want to encourage the authors to overwork the paper as it is providing interesting data for future hybrid materials development, despite the huge number of critical comments.

We want to thank the reviewer for his/her job that allows us to improve the quality of our communication. We hope that now he/she found it suitable for publication.

Round 2

Reviewer 1 Report

The manuscript ,,Using laminar nanoclays for phycocyanin and phycoerythrin stabilization and applications as new natural hybrid pigments from microalgae extraction,, has been significantly improved and therefore can be accept in its current form. 

Author Response

We would like to thank the reviewer for their comments and help, now and before. 

Reviewer 3 Report

The authors have improved the manuscript as per the reviewer's guidance; however I recommend to double-check the entire article for English editing and some stylistic and grammar mistakes before publication. There are still some mistakes (e.g. page 3, line 116: “…modified with carmic acid improves Polymmer composites properties reducing…”). Moreover, the description in the article should be done in one grammar time. The authors should standardize the entire manuscript in past or present form (not mix it).

Author Response

The reviewer is right. We have send the manuscript to a professional for the English revision as you can see in the last version. We would like to thank the reviewer for their comments and help.